# Amortized Variational Inference in Simple Hierarchical Models

**Abhinav Agrawal**
College of Information and Computer Science
Univeristy Of Massachusetts Amherst
`aagrawal@cs.umass.edu`

**Justin Domke**
College of Information and Computer Science
Univeristy Of Massachusetts Amherst
`domke@cs.umass.edu`

## Abstract

It is difficult to use subsampling with variational inference in hierarchical models since the number of local latent variables scales with the dataset. Thus, inference in hierarchical models remains a challenge at large scale. It is helpful to use a variational family with structure matching the posterior, but optimization is still slow due to the huge number of local distributions. Instead, this paper suggests an amortized approach where shared parameters simultaneously represent all local distributions. This approach is similarly accurate as using a given joint distribution (e.g., a full-rank Gaussian) but is feasible on datasets that are several orders of magnitude larger. It is also dramatically faster than using a structured variational distribution.

## 1 Introduction

Hierarchical Bayesian models are a general framework where parameters of "groups" are drawn from some shared distribution, and then observed data is drawn from a distribution specified by each group's parameters. After data is observed, the inference problem is to infer both the parameters for each group and the shared parameters. These models have proven useful in various domains [13] including hierarchical regression amd classification [12], topic models [4, 22, 3], polling [11, 24], epidemiology [23], ecology [8], psychology [37], matrix-factorization [35], and collaborative filtering [26, 33].

A proven technique for scaling variational inference (VI) to large datasets is subsampling. The idea is that if the target model has the form $p(z, y) = p(z) \prod_i p(y_i|z)$ then an unbiased gradient can be estimated while only evaluating $p(z)$ and $p(y_i|z)$ at a few $i$ [29, 16, 20, 31, 30, 36, 15].

This paper addresses hierarchical models of the form $p(\theta, z, y) = p(\theta) \prod_i p(z_i, y_i|\theta)$, where only $y$ is observed. There are two challenges. First, the number of local latent variables $z_i$ increases with the dataset, meaning the posterior distribution increases in dimensionality. Second, there is often a dependence between $z_i$ and $\theta$ which must be captured to get strong results [15, 18].

The aim of this paper is to develop a black-box variational inference scheme that can scale to large hierarchical models without losing benefits of a joint approximation. Our solution takes three steps. First, in the true posterior, the different latent variables $z_i$ are conditionally independent given $\theta$, which suggests using a variational family of the same form. We confirm this intuition by showing that for any joint variational family $q(\theta, z)$, one can define a corresponding "branch" family $q(\theta) \prod_i q(z_i|\theta)$ such that inference will be equally accurate (theorem 2). We call inference using such a family the "branch" approach.

Second, we observe that if using the branch approach, the optimal local variational parameters can be computed only from $\theta$ and local data (eq. (12)). Thus, we propose to amortize the computation of the local variational parameters by learning a network to approximately solve that optimization. We

35th Conference on Neural Information Processing Systems (NeurIPS 2021).

show that when the target distribution is symmetric over latent variables, this will be as accurate as the original joint family, assuming a sufficiently capable amortization network (claim 5).

Third, we note that in many real hierarchical models, there are many i.i.d. data generated from each local latent variable. This presents a challenge for learning an amortization network, since the full network should deal with different numbers of data points and naturally reflect the symmetry between the inputs (that is, without having to relearn the symmetry.) We propose an approach where a preliminary "feature" network processes each datum, after which they are combined with a pooling operation which forms the input for a standard network (section 6). This is closely related to the "deep sets" [39] strategy for permutation invariance.

We validate these methods on a synthetic model where exact inference is possible, and on a user-preference model for the MovieLens dataset with 162K users who make 25M ratings of different movies. At small scale (2.5K ratings), we show similar accuracy using a dense joint Gaussian, a branch distribution, or our amortized approach. At moderate scale (180K ratings), joint inference is intractable. Branch distributions gives a meaningful answer, and the amortized approach is comparable or better. At large scale (18M ratings) the amortized approach is thousands of nats better on test-likelihoods even after branch distributions were trained for almost ten times as long as the amortized approach took to converge (fig. 6).

## 2 Hierarchical Branched Distributions

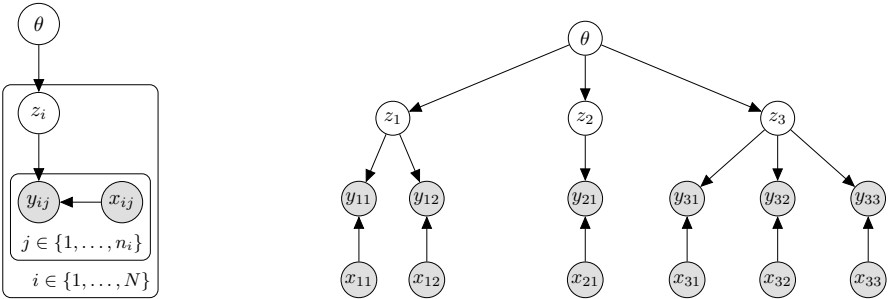

Figure 1: The graphical model for the HBDs. On the left, we have plate notation for the generic HBD from eq. (3). Note, we can have an edge from $\theta$ to $y_{ij}$ (we skip it for clarity.) On the right, we have an example model with $N = 3$.

We focus on two-level hierarchical distributions. A generic model of this type is given by

$$p(\theta, z, y | x) = p(\theta) \prod_{i=1}^{N} p(z_i | \theta) p(y_i | \theta, z_i, x_i), \tag{1}$$

where $\theta$ and $z = \{z_i\}_{i=1}^{N}$ are latent variables, $y = \{y_i\}_{i=1}^{N}$ are observations, and $x = \{x_i\}_{i=1}^{N}$ are covariates. As the visual representations of these models resemble branches, we refer them as *hierarchical branch distributions (HBDs)*.

**Symmetric.**  We call an HBD symmetric if the conditionals are symmetric, i.e., if $z_i = z_j, x_i = x_j$, and $y_i = y_j$, it implies that

$$p(z_i | \theta) = p(z_j | \theta), \text{ and}$$
$$p(y_i | \theta, z_i, x_i) = p(y_j | \theta, z_j, x_j). \tag{2}$$

**Locally i.i.d.**  Often local observations $y_i$ (and $x_i$) are a collection of conditionally i.i.d observations. Then, an HBD takes the form of

$$p(\theta, z, y | x) = p(\theta) \prod_{i=1}^{N} p(z_i | \theta) \prod_{j=1}^{n_i} p(y_{ij} | \theta, z_i, x_{ij}), \tag{3}$$

where $y_i = \{y_{ij}\}_{j=1}^{n_i}$ and $x_i = \{x_{ij}\}_{j=1}^{n_i}$ are collections of conditionally i.i.d observations and covariates; $n_i \geq 1$ is the number of observations for branch $i$.

**No local covariates.** Some applications do not involve the covariates $x_i$. In such cases, HBDs have a simplified form of

$$p(\theta, z, y) = p(\theta) \prod_{i=1}^{N} p(z_i|\theta)p(y_i|\theta, z_i). \tag{4}$$

In this paper, we will be using eq. (1) and eq. (3) to refer HBDs—the results extend easily to case where there are no local covariates. (For instance, in section 5, we amortize using $(x_i, y_i)$ as inputs. When there are no covariates, we can amortize with just $y_i$.)

## 2.1 Related Work

Bayesian inference in hierarchical models is an old problem. The most common solutions are Markov chain Monte Carlo (MCMC) and VI. A key advantage of VI is that gradients can sometimes be estimated using only a subsample of data [29]. Hoffman et al. [16] observe that inference in *hierarchical* models is still slow at large scale, since the number of parameters scales with the dataset. Instead, they assume that $\theta$ and $z_i$ from eq. (1) are in conjugate exponential families, and observe that for a mean-field variational distribution $q(\theta) \prod_i q(z_i)$, the optimal $q(z_i)$ can be calculated in closed form for fixed $q(\theta)$. This is highly scalable, though it is limited to factorized approximations and requires a conditionally conjugate target model.

A structured variational approximation like $q(\theta) \prod_i q(z_i|\theta)$ can be used which reflects the dependence of $z_i$ on $\theta$ [15, 34, 2, 18]. However, this still has scalability problems in general since the number of parameters grows in the size of the data (section 7). To the best of our knowledge, the only approach that avoids this is the framework of structured stochastic VI [15, 18], which assumes the target is conditionally conjugate, and that for a fixed $\theta$ an optimal "local" distribution $q(z_i|\theta)$ can be calculated from local data. Hoffman and Blei [15] address matrix factorization models and latent Dirichlet allocation, using Gibbs sampling to compute the local distributions. Johnson et al. [18] use amortization for conjugate models but do not consider the setting where local observations are a collection of i.i.d observations. Our approach is not strictly an instance of either of these frameworks, as we do not assume conjugacy or that amortization can exactly recover optimal local distributions [15, Eq. 7]. Still the spirit is the same, and our approach should be seen as part of this line of research.

Amortized variational approximations have been used to learn models with local variables [20, 10, 17, 5]. A particularly related instance of model learning is the Neural Statistician approach [10], where a "statistics network" learns representations of closely related datasets–the construction of this network is similar to our "pooling network" in section 6. However, in the Neural Statistician model there are no global variables $\theta$, and it is not obvious how to generalize their approach for HBDs. In contrast, our "pooling network" results from the analysis in section 5, reducing the architectural space when $\theta$ is present while retaining accuracy. Moreover, *learning* a model is strictly different from our "black-box" setting where we want to approximate the posterior of a given model, and the inference only has access to $\log p$ or $\nabla_{\theta,z} \log p$ (or their parts) [30, 21, 2, 1].

## 3 Joint approximations for HBD

When dealing with HBDs, a non-structured distribution does not scale. To see this, consider the naive VI objective—Evidence Lower Bound (ELBO, $\mathcal{L}$). Let $q_\phi^{\text{Joint}}$ be a joint variational approximation over $\theta$ and $z$; we use sans-serif font for random variables. Then,

$$\mathcal{L}\left(q_\phi^{\text{Joint}} \| p\right) = \mathbb{E}_{q_\phi^{\text{Joint}}(\theta, z)} \left[ \log \frac{p(\theta, z, y|x)}{q_\phi^{\text{Joint}}(\theta, z)} \right], \tag{5}$$

where $\phi$ are variational parameters. Usually the above expectation is not tractable and one uses a Monte-Carlo estimator. A single sample estimator is given by

$$\widehat{\mathcal{L}} = \log \frac{p(\theta, z, y|x)}{q_\phi^{\text{Joint}}(\theta, z)}, \tag{6}$$

where $(\theta, z) \sim q_\phi^{\text{Joint}}$ and $\widehat{\mathcal{L}}$ is unbiased. Since $q_\phi^{\text{Joint}}$ need not factorize, even a single estimate requires sampling all the latent variables at each step. This is problematic when there are a large number of local latent variables. One encounters the same scaling problem when taking the gradient of the ELBO. We can estimate the gradient using any of the several available estimators [30, 20, 31, 32]; however, none of them scale if $q_\phi^{\text{Joint}}$ does not factorize [16].

# 4 Branch approximations for HBD

It is easy to see that the posterior distribution of the HBD eq. (1) takes the form

$$p(\theta, z|y, x) = p(\theta|y, x) \prod_{i=1}^{N} p(z_i|\theta, y_i, x_i).$$

As such, it is natural to consider variational distributions that factorize the same way (see fig. 2.) In this section, we confirm this intuition—we start with any joint variational family which can have any dependence between $\theta$ and $z_1, \cdots, z_N$. Then, we define a corresponding "branch" family where $z_1, \cdots, z_N$ are conditionally independent given $\theta$. We show that inference using the branch family will be at least as accurate as using the joint family. We formalize the idea of branch distribution in the next definition.

**Definition 1.** Let $q_\phi^{\mathrm{Joint}}$ be any variational family with parameters $\phi$. We define $q_{v,w}^{\mathrm{Branch}}$ to be a corresponding branch family if, for all $\phi$, there exists $(v, \{w_i\}_{i=1}^{N})$, such that,

$$q_{v,w}^{\mathrm{Branch}}(\theta, z) = q_v(\theta) \prod_{i=1}^{N} q_{w_i}(z_i|\theta) = q_\phi^{\mathrm{Joint}}(\theta) \prod_{i=1}^{N} q_\phi^{\mathrm{Joint}}(z_i|\theta). \tag{7}$$

Given a joint distribution, a branch distribution can always be defined by choosing $w$ and $v_i$ as the components of $\phi$ that influence $q_\phi^{\mathrm{Joint}}(\theta)$ and $q_\phi^{\mathrm{Joint}}(z_i|\theta)$, respectively, and choosing $q_w(\theta)$ and $q_{w_i}(z_i|\theta)$ correspondingly. However, the choice is not unique (for instance, the parameterization can require transformations—different transformations can create different variants.)

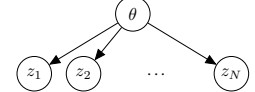

Figure 2: $q_{v,w}^{\mathrm{Branch}}$ as in eq. (7).

The idea to use $q_{v,w}^{\mathrm{Branch}}$ is natural [15, 2]. However, one might question if the branch variational family is as good as the original $q_\phi^{\mathrm{Joint}}$; in theorem 2, we establish that this is indeed true.

**Theorem 2.** Let $p$ be a HBD, and $q_\phi^{\mathrm{Joint}}(\theta, z)$ be a joint approximation family parameterized by $\phi$. Choose a corresponding branch variational family $q_{v,w}^{\mathrm{Branch}}(\theta, z)$ as in definition 1. Then,

$$\min_{v,w} KL\left(q_{v,w}^{\mathrm{Branch}}\|p\right) \leq \min_{\phi} KL\left(q_\phi^{\mathrm{Joint}}\|p\right).$$

We stress that $q_{v,w}^{\mathrm{Branch}}$ is a new variational family derived from but not identical to $q_{v,\phi}^{\mathrm{Joint}}$. Theorem 2 implies that we can optimize a branched variational family $q_{v,w}^{\mathrm{Branch}}$ without compromising the quality of approximation (see appendix D for proof.) In the following corollary, we apply theorem 2 to a joint Gaussian to show that using a branch Gaussian will be equally accurate.

**Corrolary 3.** Let $p$ be a HBD, and let $q_\phi^{\mathrm{Joint}}(\theta, z) = \mathcal{N}((\theta, z)|\mu, \Sigma)$ be a joint Gaussian approximation (with $\phi = (\mu, \Sigma)$). Choose a variational family

$$q_{v,w}^{\mathrm{Branch}}(\theta, z) = \mathcal{N}(\theta|\mu_0, \Sigma_0) \prod_{i=1}^{N} \mathcal{N}(z_i|\mu_i + A_i\theta, \ \Sigma_i)$$

with $v = (\mu_0, \Sigma_0)$ and $w_i = (\mu_i, \Sigma_i, A_i)$. Then, $\min_{v,w} KL\left(q_{v,w}^{\mathrm{Branch}}\|p\right) \leq \min_{\phi} KL\left(q_\phi^{\mathrm{Joint}}\|p\right).$

In the above corollary, the structured family $q_{v,w}^{\mathrm{Branch}}$ is chosen such that it can represent any branched Gaussian distribution. Notice, the mean of the conditional distribution is an affine function of $\theta$. This affine relationship appears naturally when you factorize the joint Gaussian over $(\theta, z)$. For more details see appendix F.

## 4.1 Subsampling in branch distributions

In this section, we show that if $p$ is an HBD and $q_{v,w}^{\mathrm{Branch}}$ is as in definition 1, we can estimate ELBO using local observations and scale better. Consider the ELBO

$$\mathcal{L}\left(q_{v,w}^{\mathrm{Branch}}\|p\right) = \mathop{\mathbb{E}}_{q_{v,w}^{\mathrm{Branch}}(\theta, z)} \left[\log \frac{p(\theta, z, y|x)}{q_{v,w}^{\mathrm{Branch}}(\theta, z)}\right]$$

$$
\begin{aligned}
&\texttt{JointELBO}(\phi, y, x) \\
&\quad \theta, z \sim q_\phi^{\text{Joint}}(\theta, z) \\
&\quad \widehat{\mathcal{L}} \leftarrow \log \frac{p(\theta, z, y|x)}{q_\phi^{\text{Joint}}(\theta, z)}
\end{aligned}
$$

(a) Estimation with $q_\phi^{\text{Joint}}$ as in eq. (6)

$$
\begin{aligned}
&\texttt{BranchELBO}(v, w, y, x) \\
&\quad \theta \sim q_v(\theta) \\
&\quad z_i \sim q_{w_i}(z_i|\theta) \text{ for } i \in \{1, \cdots, N\}. \\
&\quad \widehat{\mathcal{L}} \leftarrow \log \frac{p(\theta)}{q_v(\theta)} + \sum_{i=1}^{N} \log \frac{p(z_i, y_i|\theta, x_i)}{q_{w_i}(z_i|\theta)}
\end{aligned}
$$

(b) Estimation with $q_{v,w}^{\text{Branch}}$ as in eq. (9)

$$
\begin{aligned}
&\texttt{SubSampledBranchELBO}(v, w, y, x) \\
&\quad \theta \sim q_v(\theta) \\
&\quad B \sim \texttt{Minibatch(B)} \\
&\quad z_i \sim q_{w_i}(z_i|\theta) \text{ for } i \in B \\
&\quad \widehat{\mathcal{L}} \leftarrow \log \frac{p(\theta)}{q_v(\theta)} + \frac{N}{|B|} \sum_{i \in B} \log \frac{p(z_i, y_i|\theta, x_i)}{q_{w_i}(z_i|\theta)}
\end{aligned}
$$

(c) Estimation with $q_{v,w}^{\text{Branch}}$ as in eq. (10)

$$
\begin{aligned}
&\texttt{AmortizedSubSampledBranchELBO}(v, u, y, x) \\
&\quad \theta \sim q_v(\theta) \\
&\quad B \sim \texttt{Minibatch(B)} \\
&\quad w_i \leftarrow \texttt{net}_u(x_i, y_i) \text{ for } i \in B \\
&\quad z_i \sim q_{w_i}(z_i|\theta) \text{ for } i \in B \\
&\quad \widehat{\mathcal{L}} \leftarrow \log \frac{p(\theta)}{q_v(\theta)} + \frac{N}{|B|} \sum_{i \in B} \log \frac{p(z_i, y_i|\theta, x_i)}{q_{w_i}(z_i|\theta)}
\end{aligned}
$$

(d) Estimation with $q_{v,u}^{\text{Amort}}$ for $p$ as in eq. (2)

Figure 3: Pseudo codes for ELBO estimation with different variational methods; $w = \{w_i\}_{i=1}^{N}$, $y = \{y_i\}_{i=1}^{N}$, and $y_i = \{y_{ij}\}_{j=1}^{n_i}$ ($x$ is defined similar to $y$.) (a) Estimates ELBO for a joint approximation; (b) to (d) estimate ELBO for branch approximations; (c, d) use subsampling to estimate ELBO; (d) uses amortized conditionals; (a) to (c) work for any HBD, and (d) assumes $p$ is a symmetric HBD as in eq. (2). For models where $n_i > 1$, we use the $\texttt{net}_u$ as in fig. 4. Minibatch is some distribution over the set of possible minibatches and $|B|$ denotes the number of samples in a minibatch $B$.

$$
= \mathop{\mathbb{E}}_{q_v(\theta)} \left[ \log \frac{p(\theta)}{q_v(\theta)} \right] + \sum_{i=1}^{N} \mathop{\mathbb{E}}_{q_v(\theta)} \mathop{\mathbb{E}}_{q_{w_i}(z_i|\theta)} \left[ \log \frac{p(z_i, y_i|\theta, x_i)}{q_{w_i}(z_i|\theta)} \right]. \tag{8}
$$

Without assuming special structure (e.g. conjugacy) the above expectations will not be available in closed form. To estimate the ELBO, let $(\theta, \{z_i\}_{i=1}^{N}) \sim q_{v,w}^{\text{Branch}}$. Then, an unbiased estimator is

$$
\widehat{\mathcal{L}} = \log \frac{p(\theta)}{q_v(\theta)} + \sum_{i=1}^{N} \left[ \log \frac{p(z_i, y_i|\theta, x_i)}{q_{w_i}(z_i|\theta)} \right]. \tag{9}
$$

Unlike the joint estimator of eq. (6), one can subsample the terms in eq. (9) to create a new unbiased estimator. Let B be randomly selected minibatch of indices from $\{1, 2, \ldots, N\}$. Then,

$$
\widehat{\mathcal{L}} = \log \frac{p(\theta)}{q_v(\theta)} + \frac{N}{|\mathsf{B}|} \sum_{i \in \mathsf{B}} \left[ \log \frac{p(z_i, y_i|\theta, x_i)}{q_{w_i}(z_i|\theta)} \right], \tag{10}
$$

is another unbiased estimator of ELBO. In figs. 3b and 3c, we present the complete pseudocodes for ELBO estimation with and without subsampling in branch distributions.

Unsurprisingly, the same summation structure appears for gradients estimators of branch ELBO, allowing for efficient gradient estimation. With subsampled evaluation and training, branch distributions are immensely computationally efficient—in our experiments, we scale to models with $10^3$ *times* more latent variables by switching to branch approximations (see fig. 6).

While branch distributions are immensely more scalable than joint approximations, the number of parameters still scales as $\mathcal{O}(N)$. In the next section, we demonstrate that for symmetric HBDs, we can share parameters for the local conditionals (amortize) to allow further scalability.

## 5  Amortized branch approximations

In this section, we discuss how one can amortize the local conditionals of a branch approximations when the target HBD is symmetric (see eq. (2).) We first formally introduce the amortized branch distributions in the next definition and then justify the amortization for symmetric HBD.

**Definition 4.** Let $q_\phi^{\text{Joint}}$ be a joint approximation and let $q_{v,w}^{\text{Branch}}$ be as in definition 1. Suppose $\text{net}_u(x_i, y_i)$ is some parameterized map (with parameters $u$) from local observations $(x_i, y_i)$ to space of $w_i$. Then,

$$q_{v,u}^{\text{Amort}}(\theta, z) = q_v(\theta) \prod_{i=1}^{N} q_{\text{net}_u(x_i, y_i)}(z_i | \theta) \tag{11}$$

is a corresponding amortized branch distribution.

The idea to amortize is natural once you examine the optimization for symmetric HBDs. Consider the optimization for objective in eq. (8).

$$\max_{v,w} \mathcal{L}\left(q_{v,w}^{\text{Branch}} \| p\right) = \max_{v,w} \left[ \underset{q_v(\theta)}{\mathbb{E}} \left[ \log \frac{p(\theta)}{q_v(\theta)} \right] + \sum_{i=1}^{N} \underset{q_v(\theta)}{\mathbb{E}} \underset{q_{w_i}(z_i|\theta)}{\mathbb{E}} \left[ \log \frac{p(z_i, y_i | \theta, x_i)}{q_{w_i}(z_i|\theta)} \right] \right]$$

$$= \max_{v} \left[ \underset{q_v(\theta)}{\mathbb{E}} \left[ \log \frac{p(\theta)}{q_v(\theta)} \right] + \sum_{i=1}^{N} \max_{w_i} \underset{q_v(\theta)}{\mathbb{E}} \underset{q_{w_i}(z_i|\theta)}{\mathbb{E}} \left[ \log \frac{p(z_i, y_i | \theta, x_i)}{q_{w_i}(z_i|\theta)} \right] \right].$$

The crucial observation in the above equation is that for any given $v$, the optimal solution of inner optimization depends only on local data points $(x_i, y_i)$, i.e.,

$$w_i^* = \underset{w_i}{\text{argmax}} \; \underset{q_v(\theta)}{\mathbb{E}} \underset{q_{w_i}(z_i|\theta)}{\mathbb{E}} \left[ \log \frac{p(z_i, y_i | \theta, x_i)}{q_{w_i}(z_i|\theta)} \right]. \tag{12}$$

Now, notice that if $p$ and $q_{v,w}^{\text{Branch}}$ have symmetric conditionals, then, for each $i$, we solve the same optimization over $w_i$, just with different parameters $y_i$ and $x_i$. Thus, one could replace the optimization over $w_i$ with an optimization over a parameterized function from $(x_i, y_i)$ to the space of $w_i$. Formally, when the network $\text{net}_u$ is sufficiently capable, we make the following claim.

**Claim 5.** Let $p$ be a symmetric HBD and let $q_\phi^{\text{Joint}}$ be some joint approximation. Let $q_{v,u}^{\text{Amort}}$ be as in definition 1. Suppose that for all $v$, there exists a $u$, such that,

$$\text{net}_u(x_i, y_i) = \underset{w_i}{\text{argmax}} \; \underset{q_v(\theta)}{\mathbb{E}} \underset{q_{w_i}(z_i|\theta)}{\mathbb{E}} \left[ \log \frac{p(z_i, y_i | \theta, x_i)}{q_{w_i}(z_i|\theta)} \right]. \tag{13}$$

Then,

$$\min_{v,u} KL\left(q_{v,u}^{\text{Amort}} \| p\right) \leq \min_{\phi} KL\left(q_\phi^{\text{Joint}} \| p\right) \tag{14}$$

Note, we only amortize the conditional distribution $q_{w_i}(z_i | \theta)$ and leave $q_v(\theta)$ unchanged. In practice, of course, we do not have perfect amortization functions. The quality of the amortization depends on our ability to parameterize and optimize a powerful neural network. In other words, we make the following approximation

$$\text{net}_u(x_i, y_i) \approx \underset{w_i}{\text{argmax}} \; \underset{q_v(\theta)}{\mathbb{E}} \underset{q_{w_i}(z_i|\theta)}{\mathbb{E}} \left[ \log \frac{p(z_i, y_i | \theta, x_i)}{q_{w_i}(z_i|\theta)} \right]. \tag{15}$$

In our experiments, we found the amortized approaches work well even with moderately sized networks. Due to parameter sharing, the amortized approaches converge much faster than other alternatives (especially, true for larger models; see fig. 6 and table 2).

## 6 Amortized branch approximations for i.i.d. observations

In the previous section, we discussed how we could amortize branch approximations for symmetric HBDs. However, in some applications, the construction of amortization network $\text{net}_u$ is not as straightforward. Consider the case when we have a varying number of local i.i.d observations for each local latent variable. In this section, we highlight the problem with naive amortization for locally i.i.d HBDs, and present a simple solution to alleviate them.

Table 1: All variational families used in our experiments. $\Sigma$ denotes a generic covariance matrix and $\sigma^2$ denotes a diagonal covariance.

| Gaussian Family | $q_\phi^{\text{Joint}}$ as in eq. (5) | $q_{v,w}^{\text{Branch}}$ as in eq. (7) | $q_{v,u}^{\text{Amort}}$ as in definition 4 |
|---|---|---|---|
| Dense | $\mathcal{N}(\theta, z \mid \mu, \Sigma)$ | $\mathcal{N}(\theta \mid \mu_0, \Sigma_0) \prod_{i=1}^{N} \mathcal{N}(z_i \mid \mu_i + A_i\theta, \Sigma_i)$ | $\mathcal{N}(\theta \mid \mu_0, \Sigma_0) \prod_{i=1}^{N} \mathcal{N}(z_i \mid \mu_i + A_i\theta, \Sigma_i)$ |
| | $\phi = (\mu, \Sigma)$ | $v = (\mu_0, \Sigma_0), w_i = (\mu_i, A_i, \Sigma_i)$ | $v = (\mu_0, \Sigma_0), (\mu_i, A_i, \Sigma_i) = \texttt{net}_u(x_i, y_i)$ |
| Block Diagonal | $\mathcal{N}(\theta \mid \mu_0, \Sigma_0)\mathcal{N}(z \mid \mu_1, \Sigma_1)$ | $\mathcal{N}(\theta \mid \mu_0, \Sigma_0) \prod_{i=1}^{N} \mathcal{N}(z_i \mid \mu_i, \Sigma_i)$ | $\mathcal{N}(\theta \mid \mu_0, \Sigma_0) \prod_{i=1}^{N} \mathcal{N}(z_i \mid \mu_i, \Sigma_i)$ |
| | $\phi = (\mu_0, \mu_1, \Sigma_0, \Sigma_1)$ | $v = (\mu_0, \Sigma_0), w_i = (\mu_i, \Sigma_i)$ | $v = (\mu_0, \Sigma_0), (\mu_i, \Sigma_i) = \texttt{net}_u(x_i, y_i)$ |
| Diagonal | $\mathcal{N}(\theta, z \mid \mu, \sigma^2)$ | $\mathcal{N}(\theta \mid \mu_0, \sigma_0^2) \prod_{i=1}^{N} \mathcal{N}(z_i \mid \mu_i, \sigma_i^2)$ | $\mathcal{N}(\theta \mid \mu_0, \sigma_0^2) \prod_{i=1}^{N} \mathcal{N}(z_i \mid \mu_i, \sigma_i^2)$ |
| | $\phi = (\mu, \sigma^2)$ | $v = (\mu_0, \sigma_0^2), w_i = (\mu_i, \sigma_i^2)$ | $v = (\mu_0, \sigma_0^2), (\mu_i, \sigma_i^2) = \texttt{net}_u(x_i, y_i)$ |

Mathematically, for locally i.i.d HBDs we have $y_i = \{y_{ij}\}_{j=1}^{n_i}$ and $x_i = \{x_{ij}\}_{j=1}^{n_i}$, such that the conditional over $y_i$ factorizes as

$$p\left(y_i \mid x_i, z_i, \theta\right) = \prod_{j=1}^{n_i} p\left(y_{ij} \mid x_{ij}, z_i, \theta\right).$$

Now, if $x_i$ and $y_i$ are directly input to the amortization network $\texttt{net}_u$, the input size to the network would change for different $i$ (notice we have $n_i$ observations for $i^{\text{th}}$ local variable.) Another problem is that the optimal variational parameters are invariant to the order in which the i.i.d. observations are presented. For instance, consider two data points: $(x_i, y_i) = [(x_{i1}, y_{i1}), \ldots, (x_{in_i}, y_{in_i})]$ and $(x_i', y_i') = [(x_{in_i}, y_{in_i}), \ldots, (x_{i1}, y_{i1})]$. A naive amortization scheme will evaluate very different conditionals for these two data points because $\texttt{net}_u(x_i, y_i)$ and $\texttt{net}_u(x_i', y_i')$ will be different.

To deal with both issues: variable length input and permutation invariance, we suggest learning a "feature network" and "pooling function" based amortization network; this is reminiscent of "deep sets" [39] albeit here intended not just to enforce permutation invariance but also to deal with inputs of different sizes. Firstly, a feature network $\texttt{feat\_net}$ takes each $(x_{ij}, y_{ij})$ pair and returns a vector of features $e_j$. Secondly, a pooling function $\texttt{pool}$ takes the collection $\{e_j\}_{j=1}^{n_i}$ and achieves the two aims. First, it collapses $n_i$ feature vectors into a single fixed-sized feature $e$ (with the same dimensions as $e_j$). Second, pooling is invariant by construction to the order of observations (for example, pooling function would take a dimension-wise mean or sum across $j$.) Finally, this pooled feature vector $e$ is input to another network $\texttt{param\_net}$ that returns the final parameters $w_i$. The pseudocode for a $\texttt{net}_u$ with feature networks is available in fig. 4. In table 4, in appendix, we summarize the applicability of proposed variational methods to different HBD variants.

```
net_u(x_i, y_i)
    for j in {1, 2, ..., n_i}
        e_j ← feat_net_u(x_ij, y_ij)
    e ← pool({e_j}_{j=1}^{n_i})
    w_i ← param_net_u(e)
    return w_i
```

Figure 4: Psuedocode for $\texttt{net}_u$ for locally i.i.d symmetric HBD.

## 7 Experiments

We conduct experiments on a synthetic and a real-world problem. For each, we consider three inference methods: using a joint distribution, using a branch variational approximation, and using our amortized approach. For each method, we consider three variational approximations a completely diagonal Gaussian, a block-diagonal Gaussian (with blocks for $\theta$ and $z$) and a dense Gaussian; see fig. 5 for a visual. (For each choice of a joint distribution, the corresponding $q_{v,w}^{\text{Branch}}$ is used for the branch variational approximation and the corresponding $q_{v,u}^{\text{Amort}}$ for the amortized approach; see table 1 for details.)

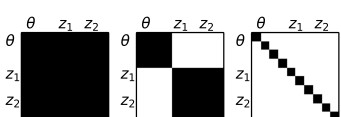

Figure 5: Visualization of dense, block-diagonal, and diagonal covariances. For each, we experiment with $q_\phi^{\text{Joint}}$, $q_{v,w}^{\text{Branch}}$, and $q_{v,u}^{\text{Amort}}$ methods.

We use reparameterized gradients [20] and optimize with Adam [19] (see appendix G for complete experimental details.) In appendix A, we discuss some of the observations we had during experimen-

tation involving parameterization, `feat-net` architecture, batch-size selection, initialization, and gradient estimators.

## 7.1 Synthetic problem

The aim of the synthetic experiment is to work with models where we have access to closed-form posterior (and the marginal likelihood.) If the variational family contains the posterior, one can expect the methods to perform close to ideal (provided we can optimize well.)

For our experiments, we use the following hierarchical regression model

$$p(\theta, z, y|x) = \mathcal{N}(\theta|0, I) \prod_{i=1}^{N} \mathcal{N}(z_i|\theta, I) \prod_{j=1}^{n_i} \mathcal{N}(y_{ij}|x_{ij}^{\top} z_i, 1), \tag{16}$$

where $\mathcal{N}$ denotes a Gaussian distribution, and $I$ is an identity matrix (see appendix G.2 for posterior and the marginal closed-form expressions.)

To demonstrate the performance of our methods, we experiment with three different problems scale (correspond to three different models with $N = 10$, 1K, and 100K.) Synthetic data is created using forward sampling for each of the scale variants independently. We avoid any test data and metrics for the synthetic problem as the log-marginal is known in closed form. The inference results are present in fig. 7 (in appendix.) In all cases, amortized distributions perform favorably when compared to branch and joint distributions.

## 7.2 MovieLens

Next, we test our method on the MovieLens25M [14], a dataset of 25 million movie ratings for over 62,000 movies, rated by 162,000 users, along with a set of features (tag relevance scores [38]) for each movie.

Purely, to make experiments more efficient on GPU hardware, we pre-process the data to drop users with more than 1,000 ratings—leaving around 20M ratings. Also, for the sake of efficiency, we PCA the movie features to reduce their dimensionality to 10. We used a train-test split such that, for each user, one-tenth of the ratings are in the test set. This gives us $\approx 18M$ ratings for training (and $\approx 2M$ ratings for testing.)

Table 3: Details for different scales of the MovieLens 25M problem.

| Scale | # of ratings | # of users | $\frac{\text{\# ratings}}{\text{\# of users}}$ |
|---|---|---|---|
| Small | 2.5K | 16 | 156.3 |
| Moderate | 180K | 1600 | 112.5 |
| Large | 18M | 159978 | 112.5 |

We use the hierarchical model

$$p(\theta, z, y|x) = \mathcal{N}(\theta|0, I) \prod_{i=1}^{N} \mathcal{N}(z_i|\mu(\theta), \Sigma(\theta)) \prod_{j=1}^{n_i} \mathcal{B}(y_{ij}|\text{sigmoid}(x_{ij}^{\top} z_i)), \tag{17}$$

Table 2: Inference results for the MovieLens25M problem. For both metrics, we draw a fresh batch of 10,000 samples from the final posterior. All values are in nats (higher is better).

| Metric | | | Final ELBO | | | Test likelihood | |
|---|---|---|---|---|---|---|---|
| $\approx$ # train ratings | | 2.5K | 180K | 18M | 2.5K | 180K | 18M |
| Methods (see table 1) | | | | | | | |
| Dense | $q_{\phi}^{\text{Joint}}$ | -1572.31 | | | -166.37 | | |
| | $q_{v,w}^{\text{Branch}}$ | -1572.39 | -1.0368e+05 | -1.1413e+07 | -166.66 | -11054.43 | -1.3046e+06 |
| | $q_{v,u}^{\text{Amort}}$ | -1572.45 | -1.0352e+05 | -1.0665e+07 | -166.64 | -10976.38 | -1.1476e+06 |
| Block | $q_{\phi}^{\text{Joint}}$ | -1579.04 | | | -167.36 | | |
| Diagonal | $q_{v,w}^{\text{Branch}}$ | -1579.05 | -1.0350e+05 | -1.1078e+07 | -166.97 | -10987.17 | -1.2538e+06 |
| | $q_{v,u}^{\text{Amort}}$ | -1579.06 | -1.0353e+05 | -1.0665e+07 | -166.96 | -10975.96 | -1.1484e+06 |
| Diagonal | $q_{\phi}^{\text{Joint}}$ | -1592.59 | | | -167.39 | | |
| | $q_{v,w}^{\text{Branch}}$ | -1592.64 | -1.0428e+05 | -1.1325e+07 | -167.31 | -10977.95 | -1.2713e+06 |
| | $q_{v,u}^{\text{Amort}}$ | -1592.64 | -1.0430e+05 | -1.0736e+07 | -167.29 | -10980.75 | -1.1497e+06 |

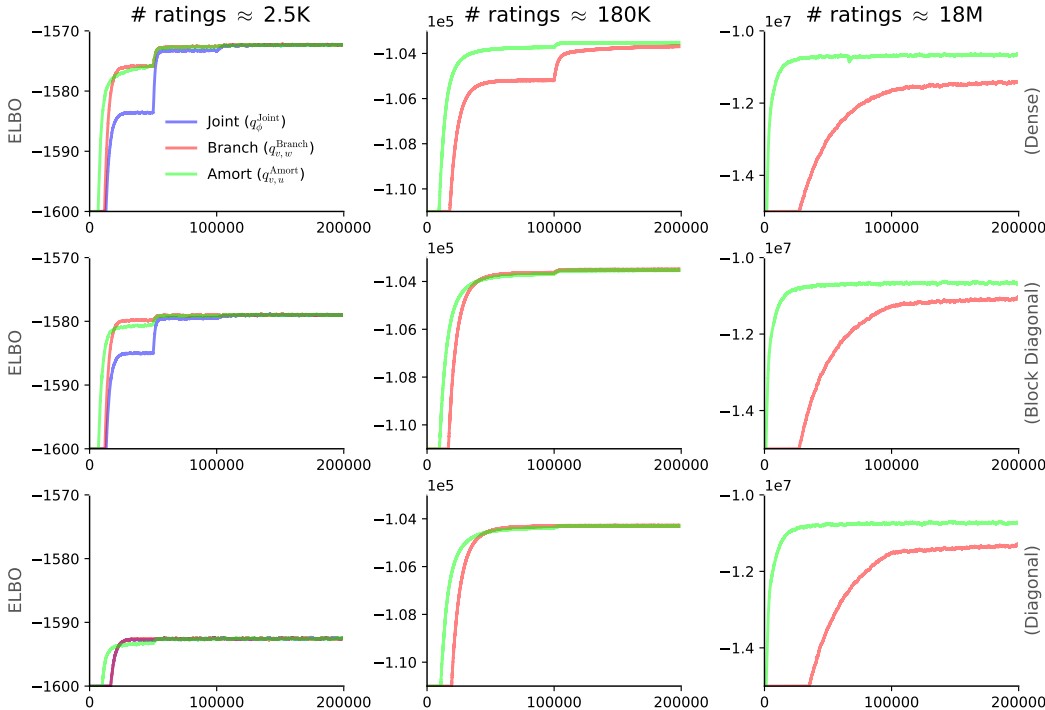

Figure 6: Training ELBO trace for the MovieLens25M problem. Top to bottom: dense, block diagonal, and diagonal Gaussian (for each, we have $q_\phi^{\text{Joint}}$, $q_{v,w}^{\text{Branch}}$, and $q_{v,u}^{\text{Amort}}$ method.) Left to right: small, moderate, and large scale of the MovieLens25M problem. For clarity, we plot the exponential moving average of the training ELBO trace with a smoothing value of 0.001. These traces correspond to the values reported in table 2.

where $\theta$ represents distribution over user preferences; for instance, $\theta$ might represent that users who like action films tend to also like thrillers but tend to dislike musicals; $z_i$ determine the user specific preference; $x_{ij}$ are the features of the $j^{\text{th}}$ movie rated by user $i$; $y_{ij}$ is the binary movie ratings; $n_i$ is the number of movies rated by user $i$, and $\mathcal{B}$ denotes a Bernoulli distribution. Here, $\mu$ and $\Sigma$ are functions of $\theta$, such that, for $\theta = [\theta_\mu, \theta_\Sigma]$, we have

$$\mu(\theta) = \theta_\mu, \qquad \text{and} \qquad \Sigma(\theta) = \text{tril}(\theta_\Sigma)^\top \text{tril}(\theta_\Sigma)$$

where $\text{tril}$ is a function that transforms an unconstrained vector into a lower-triangular positive Cholesky factor. As movie features $x_{ij} \in \mathbb{R}^{10}$, we have $\theta_\mu \in \mathbb{R}^{10}, \theta_\Sigma \in \mathbb{R}^{55}$, and $z_i \in \mathbb{R}^{10}$. Note that as $\Sigma$ depends on $\theta$, and the likelihood is Bernoulli, the model is non-conjugate.

For inference, we use the methods as described in table 1. Note, we hold the amortization network architecture constant across the scales–the number of parameters remains fixed for $q_{v,u}^{\text{Amort}}$ (for all Gaussian variants) while the number of parameters scale as $\mathcal{O}(N)$ for $q_{v,w}^{\text{Branch}}$ (see appendix F for more details.)

In fig. 6, we plot the training time ELBO trace, and in table 2, we present the final training ELBO and test likelihood values. We approximate the test likelihood $p(y^{\text{test}}|x^{\text{test}}, x, y)$ with $\mathbb{E}_q\left[p(y^{\text{test}}|x^{\text{test}}, q)\right]$, and draw a fresh batch of 10,000 samples to approximate the expectation (see appendix G for complete details.) For the smaller model, the amortized and branch approaches perform similar to the joint approach for all three variational approximations; this supports theorem 2 and claim 5. For the moderate size model, the branch and amortize approaches are very comparable to each other, while joint approaches fail to scale. For the large model (18M ratings), amortized approaches are significantly better than branch methods. We conjecture this is because parameter sharing in amortized approaches improves convergence for models where batch size is smaller compared to total iterates–true only for large model. Interestingly, the performance of amortized Dense and Block Gaussian approximation is very similar in the large and moderate setting (see $q_{v,u}^{\text{Amort}}$ results in tables 2, 7 and 8.) We conjecture this is because the posterior over the global parameters is

very concentrated for this problem. As $\theta$ behaves like a single fixed value, Block Gaussian performs just as good as the Dense approach (see appendix B for more discussion.)

## 8  Discussion

In this paper, we present structured amortized variational inference scheme that can scale to large hierarchical models without losing benefits of joint approximations. Such models are ubiquitous in social sciences, ecology, epidemiology, and other fields. Our ideas can not only inspire further research in inference but also provide a formidable baseline for applications.

## Acknowledgements and Disclosure of Funding

This material is based upon work supported in part by the National Science Foundation under Grant No. 1908577.

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
