# A    Other practical challenges

**Parameterizing covariances**    It is common to to re-parameterize covariance matrices to a vector of unconstrained parameters. As above, the typical way to do this is via a function $\mathrm{tril}$ that maps unconstrained vectors to Cholesky factors, i.e. lower-triangular matrices with positive diagonals. This can be done by simple re-arranging the components of the vector into a lower-triangular matrix, followed by applying a function to map the entries on the diagonal components to the positive numbers. In our preliminary experiments, the choice of the mapping was quite significant in terms of how difficult optimization was. Common choices like the $\exp(x)$ and $\log(\exp(x)+1)$ functions did not perform well when the outputs were close to zero. Instead, we propose to use the transformation $x \mapsto \frac{1}{2}(x + \sqrt{x^2 + 4\gamma})$, where $\gamma$ is a hyperparameter (we use $\gamma = 1$). This is based on the proximal operator for the multivariate Gaussian entropy [9, section 5]. Intuitively, when $x$ is a large positive number, this mapping returns approximately $x$, while if $x$ is a large negative number, the mapping returns approximately $-1/x$. This decays to zero more slowly than common mappings, which appears to improve numerical stability and the conditioning of the optimization.

**Feature network architecture.**    In this paper, we propose the use of a separate `feat_net` to deal with variable-length input and order invariance. In our preliminary experiments, we found that the performance improves when we concatenate the embedding $e_j$ in the fig. 4 with it's dimension-wise square before sending it to the pooling function `pool`. We hypothesis that this is because the embeddings act as learnable statistics, and using the elementwise square directly provides useful information to `param_net`.

**Batch size selection**    For the small scale problems (both synthetic and MovieLens), we do not sub-sample data; this is to maintain a fair comparison to joint approaches that do not support subsampling. For moderate and large scales, we select the batch size for the branch methods based on the following rule of thumb: we increase the batch size such that $\frac{|B|}{T^{1.18}}$ is roughly maximized. This rule-of-thumb captures the following intuition. Suppose batch size $|B|$ takes time $T$ per iteration. If the time taken per iteration for batchsize $|2B|$ is less than $1.8T$, then we should use $2B$. Of course, we roughly maximize $\frac{|B|}{T^{1.18}}$ for computational ease. We use the same batch size as the branch methods for the amortized methods as we found that amortized methods were much more robust to the choice of batchsize. We use $|B| = 200$ for moderate scale and $|B| = 400$ for large scale.

**Initialization**    We initialize the neural network parameters using a truncated normal distribution with zero mean and standard deviation equal to $\sqrt{1/\mathrm{fan\_in}}$, where $\mathrm{fan\_in}$ is the number of inputs to the layer [25]. We initialize the final output layer of the `param-net` with a zero mean Gaussian with a standard deviation of 0.001 [1]. This ensures an almost standard normal initialization for the local conditional $q_{\mathtt{net}_u(x_i,y_i)}(z_i|\theta)$.

**Gradient Calculation**    In our preliminary experiments, we found sticking the landing gradient [32, STL] to be less stable. STL requires a re-evaluation of the density which in turn requires a matrix inversion (for the Cholesky factor); this matrix inversion was sometimes prone to numerical precision errors. Instead, we found the regular gradient, also called the total gradient in [32], to be numerically robust as it can be evaluated without a matrix inversion. This is done by simultaneously sampling and evaluating the density in much the same as done in normlaizing flows [31, 27]. We use the total gradient for all our experiments.

# B    General trade-offs

The amortized approach proposed in section 5 is only applicable for symmetric models. In ta-ble 4, we summarize the applicability of all the methods we discuss in this paper. In our pre-liminary experiments, for amortized approaches, the performance improved when we increased the number of layers in the neural networks or increased the length of the embeddings in $\mathtt{net}_u$.

Table 4: Summary of method applicability.

| Models | $q_{v,w}^{\mathrm{Branch}}$ (definition 1) | $q_{v,u}^{\mathrm{Amort}}$ (definition 4) | $q_{v,u}^{\mathrm{Amort}}$ w/ $\mathtt{feat\_net}_u$ ($\mathtt{net}_u$ as in fig. 4) |
|---|---|---|---|
| HBD        (eq. (1)) | ✓ | ✗ | ✗ |
| Symmetric HBD (eq. (2)) | ✓ | ✓ | ✓ |
| Locally i.i.d. Symmetric HBD (eq. (3)) | ✓ | ✗ | ✓ |

However, we make no serious efforts to find the optimal architecture. In fact, we use the same

architecture for all our experiments, across the scales. We believe the performance on a particular task can be further improved by carefully curating the neural architecture. Note that there are no architecture choices in joint or branched approaches. We also did not optimize our choice of number of samples to drawn from $q$ to estimate ELBO. This forms the second source of stochasticity and using more samples can help reduce the variance [36]. We use 10 copies for all our experiments.

A particularly interesting case arises when the number of local latent variables ($N$) is very large. In such scenarios, the true posterior $p(\theta|x, y)$ can be too concentrated. As the randomness in $\theta$ is very low, we might not gain any significant benefits from conditioning on $\theta$—as $\theta$ reduces to a fixed quantity, Dense Gaussian will work just well as the Block Gaussian (see tables 2, 7 and 8). In practice, it is hard to know this apriori; in fact, our scalable approaches allow for such analysis on large scale model.

## C  Warm up to Proof for Theorem

**Proposition 6.** We know that $KL\left(q\|p\right)$ is jointly convex in the pair $(q, p)$ [7, Theorem 2.7.2]. Suppose $q_\eta$ and $p_\eta$ are valid probability distributions for each value of random variable $\eta$. Then,

$$\mathbb{E}_\eta KL\left(q_\eta\|p_\eta\right) \geq KL\left(\mathbb{E}_\eta[q_\eta]\middle\|\mathbb{E}_\eta[p_\eta]\right). \tag{18}$$

Let us offer some explanation for why the above proposition holds. Jensen's inqequality states that if $y$ is a vector-valued random variable and $f(y)$ is convex then,

$$\mathbb{E}_y f(\mathsf{y}) \geq f(\mathbb{E}_y[\mathsf{y}]). \tag{19}$$

One can extend the Jensen's inequality when $y(\eta)$ is a vector-valued function of random variable $\eta$ and $f(y)$ is convex. Then,

$$\mathbb{E}_\eta f(y(\mathsf{\eta})) = \mathbb{E}_y f(\mathsf{y}) \geq f(\mathbb{E}_y[\mathsf{y}]) = f(\mathbb{E}_\eta[y(\mathsf{\eta})]). \tag{20}$$

To see why proposition 6 follows from eq. (20), we need to extend it to two variables; consider the case when $y(\eta)$ and $x(\eta)$ are vector-valued functions, and $f(y, x)$ is jointly convex in the pair $(y, x)$. Then,

$$\mathbb{E}_\eta f(y(\mathsf{\eta}), x(\mathsf{\eta})) \geq f(\mathbb{E}_\eta[(y(\mathsf{\eta}), x(\mathsf{\eta}))]). \tag{21}$$

The proposition simply substitutes $f$ with $KL$, $y$ with $q$, and $x$ with $p$ in eq. (21).

## D  Proof for Theorem

**Theorem** (Repeated). Let $p$ be a HBD, and $q_\phi^{\text{Joint}}(\theta, z)$ be a joint approximation family parameterized by $\phi$. Choose a corresponding branch variational family $q_{v,w}^{\text{Branch}}(\theta, z)$ as in definition 1. Then,

$$\min_{v,w} KL\left(q_{v,w}^{\text{Branch}}\|p\right) \leq \min_\phi KL\left(q_\phi^{\text{Joint}}\|p\right).$$

*Proof.* Construct a new distribution $q'_\phi$ such that

$$q'_\phi(\theta, z) = q_\phi^{\text{Joint}}(\theta) \prod_i q_\phi^{\text{Joint}}(z_i|\theta). \tag{22}$$

Then, note that $z_i$ are conditionally independent in $q'_\phi$, such that,

$$q'_\phi(z_i|\theta, z_{<i}) = q'_\phi(z_i|\theta). \tag{23}$$

From chain rule of KL-divergence, we have

$$KL\left(q_\phi^{\text{Joint}}(\mathsf{\theta}, \mathsf{z})\|p(\mathsf{\theta}, \mathsf{z}|x, y)\right) = KL\left(q_\phi^{\text{Joint}}(\mathsf{\theta})\|p(\mathsf{\theta}|x, y)\right)$$
$$+ \sum_i KL\left(q_\phi^{\text{Joint}}(\mathsf{z}_i|\mathsf{z}_{<i}, \mathsf{\theta})\|p(\mathsf{z}_i|\mathsf{z}_{<i}, \mathsf{\theta}, x, y)\right), \text{ and}$$

$$KL\left(q'_\phi(\mathsf{\theta}, \mathsf{z})\middle\|p(\mathsf{\theta}, \mathsf{z}|x, y)\right) = KL\left(q_\phi^{\text{Joint}}(\mathsf{\theta})\|p(\mathsf{\theta}|x, y)\right)$$
$$+ \sum_i KL\left(q'_\phi(\mathsf{z}_i|\mathsf{z}_{<i}, \mathsf{\theta})\middle\|p(\mathsf{z}_i|\mathsf{z}_{<i}, \mathsf{\theta}, x, y)\right).$$

Consider any arbitrary summand term. We have that

$$KL\left(q_\phi^{\text{Joint}}(\mathsf{z}_i|\mathsf{z}_{<i},\theta)\|p(\mathsf{z}_i|\mathsf{z}_{<i},\theta,x,y)\right)$$

$$\overset{(a)}{=} KL\left(q_\phi^{\text{Joint}}(\mathsf{z}_i|\mathsf{z}_{<i},\theta)\|p(\mathsf{z}_i|\theta,x_i,y_i)\right)$$

$$\overset{(b)}{=} \underset{\theta\sim q_\phi^{\text{Joint}}(\theta)}{\mathbb{E}}\ \underset{z_{<i}\sim q_\phi^{\text{Joint}}(\mathsf{z}_{<i}|\theta)}{\mathbb{E}}\left[KL\left(q_\phi^{\text{Joint}}(\mathsf{z}_i|\mathsf{z}_{<i},\theta)\|p(\mathsf{z}_i|\theta,x_i,y_i)\right)\right]$$

$$\overset{(c)}{\geq} \underset{\theta\sim q_\phi^{\text{Joint}}(\theta)}{\mathbb{E}}\left[KL\left(\underset{z_{<i}\sim q_\phi^{\text{Joint}}(\mathsf{z}_{<i}|\theta)}{\mathbb{E}}[q_\phi^{\text{Joint}}(\mathsf{z}_i|\mathsf{z}_{<i},\theta)]\Bigg\|\underset{z_{<i}\sim q_\phi^{\text{Joint}}(\mathsf{z}_{<i}|\theta)}{\mathbb{E}}[p(\mathsf{z}_i|\theta,x_i,y_i)]\right)\right]$$

$$\overset{(d)}{=} \underset{\theta\sim q_\phi^{\text{Joint}}(\theta)}{\mathbb{E}}\left[KL\left(q_\phi^{\text{Joint}}(\mathsf{z}_i|\theta)\|p(\mathsf{z}_i|\theta,x_i,y_i)\right)\right]$$

$$= KL\left(q_\phi^{\text{Joint}}(\mathsf{z}_i|\theta)\|p(\mathsf{z}_i|\theta,x_i,y_i)\right)$$

$$= KL\left(q'_\phi(\mathsf{z}_i|\theta)\|p(\mathsf{z}_i|\theta,x_i,y_i)\right)$$

$$\overset{(e)}{=} KL\left(q'_\phi(\mathsf{z}_i|\theta,\mathsf{z}_{<i})\|p(\mathsf{z}_i|\theta,\mathsf{z}_{<i},x_i,y_i)\right),$$

where (a)[1] follows from HBD structure; (b)[1] follows from definition of conditional KL divergence; (c) follows from convexity of KL divergence and Jensen's inequality (substitute $\eta \to z_{<i}$ in proposition 6 and eq. (18)); (d) follows from marginalization, and (e) follows from the conditional independence of $q'$ and $p$. Summing the above result over $i$ gives that

$$KL\left(q'_\phi\|p\right) \leq KL\left(q_\phi^{\text{Joint}}\|p\right). \tag{24}$$

Now, from definition 1, we know that for every $\phi$, there exists a corresponding $(v,w)$ such that $q_{v,w}^{\text{Branch}} = q'_\phi$. Let $\phi^* = \operatorname{argmin}_\phi KL(q_\phi^{\text{Joint}}\|p)$. Then, there exists some $q_{v,w}^{\text{Branch}} = q'_{\phi^*}$. Then, it follows that

$$\min_{v,w} KL\left(q_{v,w}^{\text{Branch}}\|p\right) \leq KL\left(q'_{\phi^*}\|p\right) \leq KL\left(q_{\phi^*}^{\text{Joint}}\|p\right) = \min_\phi KL\left(q_\phi^{\text{Joint}}\|p\right). \tag{25}$$

$\square$

# E    Proof for Claim

**Claim** (Repeated). Let $p$ be a symmetric HBD and let $q_\phi^{\text{Joint}}$ be some joint approximation. Let $q_{v,u}^{\text{Amort}}$ be as in definition 1. Suppose that for all $v$, there exists a $u$, such that,

$$\mathtt{net}_u(x_i,y_i) = \operatorname*{argmax}_{w_i}\ \underset{q_v(\theta)}{\mathbb{E}}\ \underset{q_{w_i}(\mathsf{z}_i|\theta)}{\mathbb{E}}\left[\log\frac{p\left(\mathsf{z}_i,y_i|\theta,x_i\right)}{q_{w_i}\left(\mathsf{z}_i|\theta\right)}\right]. \tag{26}$$

Then,

$$\min_{v,u} KL\left(q_{v,u}^{\text{Amort}}\|p\right) \leq \min_\phi KL\left(q_\phi^{\text{Joint}}\|p\right) \tag{27}$$

*Proof.* Consider the optimization for $q_{v,w}^{\text{Branch}}$. We have

$$\max_{v,w}\mathcal{L}\left(q_{v,w}^{\text{Branch}}\|p\right) = \max_{v,w}\left[\underset{q_v(\theta)}{\mathbb{E}}\left[\log\frac{p\left(\theta\right)}{q_v\left(\theta\right)}\right] + \sum_{i=1}^N \underset{q_v(\theta)}{\mathbb{E}}\ \underset{q_{w_i}(\mathsf{z}_i|\theta)}{\mathbb{E}}\left[\log\frac{p\left(\mathsf{z}_i,y_i|\theta,x_i\right)}{q_{w_i}\left(\mathsf{z}_i|\theta\right)}\right]\right]$$

$$= \max_v\left[\underset{q_v(\theta)}{\mathbb{E}}\left[\log\frac{p\left(\theta\right)}{q_v\left(\theta\right)}\right] + \sum_{i=1}^N \max_{w_i}\ \underset{q_v(\theta)}{\mathbb{E}}\ \underset{q_{w_i}(\mathsf{z}_i|\theta)}{\mathbb{E}}\left[\log\frac{p\left(\mathsf{z}_i,y_i|\theta,x_i\right)}{q_{w_i}\left(\mathsf{z}_i|\theta\right)}\right]\right]$$

$$\overset{(f)}{=} \max_v\left[\underset{q_v(\theta)}{\mathbb{E}}\left[\log\frac{p\left(\theta\right)}{q_v\left(\theta\right)}\right] + \sum_{i=1}^N \underset{q_v(\theta)}{\mathbb{E}}\ \underset{q_{\mathtt{net}_u(x_i,y_i)}(\mathsf{z}_i|\theta)}{\mathbb{E}}\left[\log\frac{p\left(\mathsf{z}_i,y_i|\theta,x_i\right)}{q_{\mathtt{net}_u(x_i,y_i)}\left(\mathsf{z}_i|\theta\right)}\right]\right]$$

---

[1]Note that KL-divergence on the right hand side of (a) is a conditional KL divergence; however, KL-divergence on the right hand side of (b) is not a conditional KL, but a regular KL-divergence between two distributions that are conditioned on some value $\theta$ and $z_{<i}$.

$$\leq \max_u \max_v \left[ \mathbb{E}_{q_v(\theta)} \left[ \log \frac{p(\theta)}{q_v(\theta)} \right] + \sum_{i=1}^{N} \mathbb{E}_{q_v(\theta)} \mathbb{E}_{q_{\text{net}_u(x_i,y_i)}(z_i|\theta)} \left[ \log \frac{p(z_i, y_i | \theta, x_i)}{q_{\text{net}_u(x_i,y_i)}(z_i|\theta)} \right] \right]$$

$$= \max_u \max_v \mathcal{L}\left(q_{v,u}^{\text{Amort}} \| p\right),$$

where (f) follows from the assumption in the claim. Now, from the ELBO decomposition equation, we have

$$\log p(y|x) = \mathcal{L}(q\|p) + KL(q\|p). \tag{28}$$

Therefore, we have

$$\min_u \min_v KL\left(q_{v,u}^{\text{Amort}} \| p\right) \leq \min_{v,w} KL\left(q_{v,w}^{\text{Branch}} \| p\right) \tag{29}$$

From theorem 2, we get the desired result.

$$\min_u \min_v KL\left(q_{v,u}^{\text{Amort}} \| p\right) \leq \min_{v,w} KL\left(q_{v,w}^{\text{Branch}} \| p\right) \leq \min_\phi KL\left(q_\phi^{\text{Joint}} \| p\right) \tag{30}$$

$\square$

## F  Derivation for Branch Gaussian

Let $q_\phi^{\text{Joint}}(\theta, z) = \mathcal{N}((\theta, z) | \mu, \Sigma)$ be the joint Gaussian approximation as in corrolary 3. Further, let $(\mu, \Sigma)$ be defined as

$$\mu = \begin{bmatrix} \mu_\theta \\ \mu_{z_1} \\ \vdots \\ \mu_{z_N} \end{bmatrix} \quad \text{and} \quad \Sigma = \begin{bmatrix} \Sigma_\theta & \Sigma_{\theta z_1} & \dots & \Sigma_{\theta z_N} \\ \Sigma_{\theta z_1}^\top & \Sigma_{z_1} & \dots & \Sigma_{z_1 z_N} \\ \vdots & \vdots & \ddots & \vdots \\ \Sigma_{\theta z_n}^\top & \Sigma_{z_1 z_N}^\top & \dots & \Sigma_{z_N} \end{bmatrix}. \tag{31}$$

Then, from the properties of the multivariate Gaussian [28]

$$q_\phi^{\text{Joint}}(z_i|\theta) = \mathcal{N}(z_i | \mu_{z_i|\theta}, \Sigma_{z_i|\theta}), \text{ where} \tag{32}$$

$$\mu_{z_i|\theta} = \mu_{z_i} + \Sigma_{\theta z_i}^\top \Sigma_\theta^{-1} (\theta - \mu_\theta), \text{ and} \tag{33}$$

$$\Sigma_{z_i|\theta} = \Sigma_{z_i z_i} - \Sigma_{\theta z_i}^\top \Sigma_\theta^{-1} \Sigma_{\theta z_i}. \tag{34}$$

Now, to parameterize a corresponding $q_{v,w}^{\text{Branch}}$, we use the $(\mu_i, \Sigma_i, A_i)$, such that,

$$q_{v,w}^{\text{Branch}}(z_i|\theta) = \mathcal{N}(z_i | \mu_i + A_i\theta, \Sigma_i). \tag{35}$$

## G  Experimental Details

**Architectural Details**  We use the architecture as reported in table 5 for all our amortized approaches. In addition to using $e_j$ as detailed in fig. 4, we concatenate the elementwise square before sending it to `param-net`. Thus, the input to `param-net` is not 128 dimensional but 256 dimensional. Further, we use mean as the `pool` function.

**Compute Resources**  We use JAX [6] to implement our methods. We trained using Nvidia 2080ti-12GB. All methods finished training within 4 hours. Branch approaches were at an average twice as fast as amortized variants.

**Step-size drop**  We use Adam [19] for training with an initial step-size of 0.001 (and default values for other hyperparameters.) In preliminary experiments, we found that dropping the step-size improves the performance. Starting from 0.001, we drop the step to one-tenth of it's value after a predetermined number of steps. For small scale experiments, we drop a total of three times after every 50,000 iterations (we train for 200,000 iterations.) For moderate and large scale models, we drop once after 100,000 iterations.

Table 5: Architecture details for $\text{net}_u$. Each fully-connected layer is followed by leaky-ReLU baring the last layer.

| Network | Layer Skeleton |
|---|---|
| feat-net | 64, 64, 64, 128 |
| param-net | 256, 256, 256 |

### G.1  Movielens

Table 7: This table has the extended results for the Movilens25M Dataset. All values are in nats. Higher is better.

| ≈ # ratings Methods | | Test LL 2.5K | 180K | 18M | Train LL 2.5K | 180K | 18M | Train ELBO 2.5K | 180K | 18M |
|---|---|---|---|---|---|---|---|---|---|---|
| Dense | $q_\phi^{\text{Joint}}$ | -166.37 | | | -1373.97 | | | -1572.31 | | |
| | $q_{v,w}^{\text{Branch}}$ | -166.66 | -11054.43 | -1.3046e+06 | -1374.20 | -95731.42 | -1.0315e+07 | -1572.39 | -1.0368e+05 | -1.1413e+07 |
| | $q_{v,u}^{\text{Amort}}$ | -166.64 | -10976.38 | -1.1476e+06 | -1374.27 | -95980.37 | -1.0027e+07 | -1572.45 | -1.0352e+05 | -1.0665e+07 |
| Block Diagonal | $q_\phi^{\text{Joint}}$ | -167.36 | | | -1375.56 | | | -1579.04 | | |
| | $q_{v,w}^{\text{Branch}}$ | -166.97 | -10987.17 | -1.2538e+06 | -1375.71 | -95891.42 | -1.0399e+07 | -1579.05 | -1.0350e+05 | -1.1078e+07 |
| | $q_{v,u}^{\text{Amort}}$ | -166.96 | -10975.96 | -1.1484e+06 | -1375.71 | -95962.56 | -1.0027e+07 | -1579.06 | -1.0353e+05 | -1.0665e+07 |
| Diagonal | $q_\phi^{\text{Joint}}$ | -167.39 | | | -1377.25 | | | -1592.59 | | |
| | $q_{v,w}^{\text{Branch}}$ | -167.31 | -10977.95 | -1.2713e+06 | -1377.19 | -96414.40 | -1.0709e+07 | -1592.64 | -1.0428e+05 | -1.1325e+07 |
| | $q_{v,u}^{\text{Amort}}$ | -167.29 | -10980.75 | -1.1497e+06 | -1377.20 | -96467.88 | -1.0068e+07 | -1592.64 | -1.0430e+05 | -1.0736e+07 |

Table 8: This table has the extended results for the Movilens25M Dataset. It has the same results as in table 7; however, the values are divided by the number of ratings.

| Methods | ≈ # ratings | Test LL 2.5K | 180K | 18M | Train LL 2.5K | 180K | 18M | Train ELBO 2.5K | 180K | 18M |
|---|---|---|---|---|---|---|---|---|---|---|
| Dense | $q_\phi^{\text{Joint}}$ | -0.5717 | | | -0.5108 | | | -0.5845 | | |
| | $q_{v,w}^{\text{Branch}}$ | -0.5727 | -0.5640 | -0.6486 | -0.5109 | -0.5224 | -0.5492 | -0.5845 | -0.5658 | -0.6077 |
| | $q_{v,u}^{\text{Amort}}$ | -0.5726 | -0.5600 | -0.5705 | -0.5109 | -0.5238 | -0.5339 | -0.5846 | -0.5649 | -0.5678 |
| Block Diagonal | $q_\phi^{\text{Joint}}$ | -0.5751 | | | -0.5114 | | | -0.5870 | | |
| | $q_{v,w}^{\text{Branch}}$ | -0.5738 | -0.5606 | -0.6233 | -0.5114 | -0.5233 | -0.5537 | -0.5870 | -0.5648 | -0.5898 |
| | $q_{v,u}^{\text{Amort}}$ | -0.5738 | -0.5600 | -0.5709 | -0.5114 | -0.5237 | -0.5339 | -0.5870 | -0.5650 | -0.5678 |
| Diagonal | $q_\phi^{\text{Joint}}$ | -0.5752 | | | -0.5120 | | | -0.5920 | | |
| | $q_{v,w}^{\text{Branch}}$ | -0.5749 | -0.5601 | -0.6320 | -0.5120 | -0.5261 | -0.5702 | -0.5921 | -0.5691 | -0.6029 |
| | $q_{v,u}^{\text{Amort}}$ | -0.5749 | -0.5602 | -0.5716 | -0.5120 | -0.5264 | -0.5360 | -0.5921 | -0.5691 | -0.5716 |

**Feature Dimensions**   We reduce the movie feature dimensionality to 10 using PCA. This is done with branch approaches in focus as the number of features for dense branch Gaussian scale as $\mathcal{O}(ND^3)$, where $D$ is the dimensionality of the movie features. Note, that the number of features for amortized approaches is independent of $N$ allowing for better scalability.

Table 6: Metrics used for evaluation. We use K = 10,000 samples from the posterior. Here, $(z^k, \theta^k) \sim q(z, \theta | x^{\text{train}}, y^{\text{train}})$.

| Metric | Expression |
|---|---|
| Test likelihood | $\log \frac{1}{K} \sum_k p(y^{\text{test}} | x^{\text{test}}, z^k, \theta^k)$ |
| Train likelihood | $\log \frac{1}{K} \sum_k \frac{p(y^{\text{train}}, z^k, \theta^k | x^{\text{train}})}{q(z^k, \theta^k | x^{\text{train}}, y^{\text{train}})}$ |
| Train ELBO | $\frac{1}{K} \sum_k \log \frac{p(y^{\text{train}}, z^k, \theta^k | x^{\text{train}})}{q(z^k, \theta^k | x^{\text{train}}, y^{\text{train}})}$ |

**Metrics**   We use three metrics for performance evaluation—test likelihood, train likelihood, and train ELBO. Details of the expressions are presented in table 6. We draw a batch of fresh 10,000 samples from the posterior to estimate each metric. Of course, the evaluated expressions are just approximation to the true value. In table 7 and table 8 we present the extended results. In table 8 we present the same values but normalized by the number of ratings in the dataset.

**Preprocess**   Movielens25M originally uses a 5 point ratings system. To get binary ratings, we map ratings greater than 3 points to 1 and less than and equal to 3 to 0.

### G.2   Synthetic problem

**Details of the model**   We use the hierarchical regression model

$$p(\theta, z, y | x) = \mathcal{N}(\theta | 0, I) \prod_{i=1}^{N} \mathcal{N}(z_i | \theta, I) \prod_{j=1}^{n_i} \mathcal{N}(y_{ij} | x_{ij}^\top z_i, 1)$$

for synthetic experiments. For simplicity, we use $n_i = 100$ for all $i$; we vary $N$ to create different scale variants—we use $N = 10$ for small scale, $N = 1000$ for moderate scale, and $N = 100000$ for large scale experiments; we set $x_{ij} \in \mathbb{R}^{10}$ and thus $\theta \in \mathbb{R}^{10}$ and $z_i \in \mathbb{R}^{10}$; $y_{ij} \in \mathbb{R}$.

**Details of closed-form expressions**  In the following expressions, $\theta \in \mathbb{R}^D, z_i \in \mathbb{R}^D, x_i \in \mathbb{R}^{n_i \times D}$, $y_i \in \mathbb{R}^{n_i}, y \in \mathbb{R}^{(\sum_{i=1}^{N} n_i)}$, and $I_M$ is an $M \times M$ identity matrix.

**Expression for posterior**

$$p(\theta|x,y) = \mathcal{N}\left(\left[I_D + \sum_i^N x_i^\top (I_{n_i} + x_i x_i^\top)^{-1} x_i\right]^{-1} \left[\sum_i^N x_i^\top (I_{n_i} + x_i x_i^\top)^{(-1)} y_i\right],\right.$$

$$\left.\left[I_D + \sum_i^N x_i^\top (I_{n_i} + x_i x_i^\top)^{-1} x_i\right]^{-1}\right)$$

$$p(z_i|\theta, x_i, y_i) = \mathcal{N}([I_D + x_i^\top x_i]^{-1}[x_i^\top y_i + \theta], [I_D + x_i^\top x_i]^{-1})$$

**Expression for marginal likelihood**

$$p(y|x) = \mathcal{N}\left(0, \begin{bmatrix} I_{n_1} + 2x_1 x_1^\top & \dots & x_N x_1^\top \\ \vdots & \ddots & \vdots \\ x_N x_1^\top & \dots & I_{n_N} + 2x_N x_N^\top \end{bmatrix}\right)$$

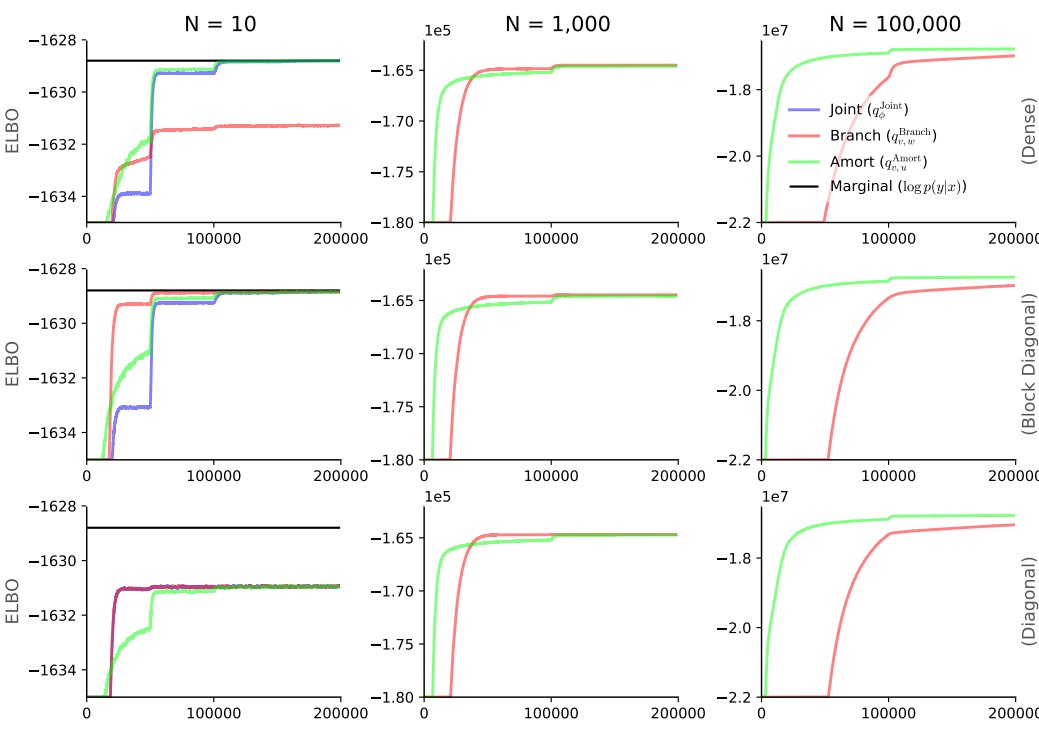

Figure 7: Training ELBO trace for the synthetic problem. Top to bottom: dense, block diagonal, and diagonal Gaussian (for each, we have $q_\phi^{\text{Joint}}$, $q_{v,w}^{\text{Branch}}$, and $q_{v,u}^{\text{Amort}}$ method.) Left to right: small, moderate, and large scale of the synthetic problem. For clarity, we plot the exponential moving average of the training ELBO trace with a smoothing value of 0.001. For the small setting, we also plot the true log-marginal $\log p(y|x)$ for reference (black horizontal line): ELBO for dense approach is exactly same as the log-marginal, it's slightly lower for block, and is much less for the diagonal (see first column.) Note, calculating the log-marginal was computationally prohibitive for the moderate and large setting.