# OpenReview forum: "Amortized Variational Inference for Simple Hierarchical Models"
_NeurIPS.cc/2021/Conference — NeurIPS 2021 Poster_

### Official Review · Reviewer_k4wz · 2021-07-09

**Rating:** 7
**Confidence:** 3

**Summary:**

The paper deals with scalable approximate inference for hierarchical models and proposes a framework for hierarchical approximations that exhibit the same factorization as the exact posterior of the hierarchical model. This efficiently enables subsampling the data when evaluating the evidence lower bound and its gradients. The authors further propose a pooling-based strategy for amortized inference to avoid the usual linear scaling in the number of approximation parameters.  The paper is concluded with a set of numerical experiments with both synthetic and real data demonstrating the merits of the proposed framework.




**Main Review:**

The paper proposes a novel (to the best of my knowledge) and elegant solution to an important problem and I think the proposed methods will be of general interest to the approximate inference community.

All technical details in the paper appear to be correct and the claims in the paper are theoretically motivated and reasonably justified using empirical experiments. However, the empirical experiments could be further improved by

- The results for the synthetic experiment are only compared using the lower bounds. Since the target distribution is Gaussian, why not 1) compute and plot the true log marginal likelihood in Figure 7 in the appendix for comparison 2) compute and evaluate the error on the estimated moments/parameters for completeness since the true quantities are available.

- Use replications and add error bars to all experiments

Overall, the paper is well-written and well-structured and I really enjoyed reading it. However, I do think the notation can be slightly improved. For example,

- I am a little bit puzzled by the exact meaning of the definition in Eq. (2). Perhaps, this is caused by the usual casual notation in probability, where there is no distinction in the notation for a random variable and the values it takes. Can this be rephrased or made more precise?

- Along the same lines, in eq. (6), (8)-(10) it would be a nice service to the reader if the authors would distinguish between the ranom variables and samples of the random variables. For example, use \theta^s or \hat{theta} to represent a sample of the random variable \theta and the same for the other variables.

- I think there may be missing a minimization operator on the left hand side of eq. (14) since currently, the left hand side is a function of v (and the existence of a uniform bound seems unlikely). I noticed that \min_v is also present in the proof in the appendix, so it is most likely just a typo.


Other comments:

- the xlabels in fig 6 are missing.

 - extra parenthesis in eq. (16)

- It's nice that the authors included a section describing some of the practical challenges and considerations.




**Time Spent Reviewing:**

2

---

> ### Author Response · Authors · 2021-08-10
> **We thank the reviewer for general comments; we will cover the typos and offer explanation for more specific concerns below.**
>
>
> # Synthetic experiment results
>
> Thank you for bringing up the concern about using true log-marginal as a diagnostic tool for the synthetic experiments; we even mention this in L225; however, we realized that for the large scale (with N = 100K, the third column in Fig 7, in appendix) it is computationally prohibitive to calculate the log density $\log p(y \vert x)$. To see this, note $y \in R^{10^{8}}$. Calculating the true marginal likelihood, for given $y$, requires working with a $10^8 \times 10^8$ covariance matrix (see section F.2 for expressions). We further simplified this calculation using the form in section F.2 by decomposing the covariance matrix as a sum of block-diagonal and a dense matrix; however, it was still computationally prohibitive.
>
> We can certainly do this calculation for the smaller scales but left it out for consistency. If the reviewers suggest, we would be happy to add the log-marginal values for the small and moderate scale models where we can calculate it. For those cases, we found the gap to be fairly small (not surprising as the Gaussian model class contains the true posterior.)
>
> Nevertheless, we believe this hints that our approaches truly scale to large models.
>
> # Replication
>
> We agree that all the experiments will benefit from replication. We do want to mention that in our initial analysis, the inherent stochasticity in our setup was minimal (partly due to the observations made in Section A on initialization and parameterization); further, the computation needs for a full-scale replication of all the results were substantial.
> To remedy this, we propose to replicate Table 2 over several trials and add the discussion to the appendix.
>
> # Definition in Eq. 2 and the random variable notation
>
> We use the sans-serif font for referring to random variables (we make a note of this in L90). In retrospect, we should have highlighted the notation more explicitly. We welcome suggestions on the change of notation: should we change it to bold sans-serif?
>
> For the comments on the symmetric definition, please refer to the response to the reviewer F7hF. The gist is that for a general hierarchical branch model, conditionals may not necessarily be in the same class of distribution (note, for symmetric HBD, the conditionals are in the same class of distributions). In doing this, we employ a common abuse of notation where $p(z)$ denotes the distribution of the random variables $z$ and the functional form that $p$ takes changes with the variable. As mentioned in response to reviewer F7hF, We will make this abuse of notation abundantly clear in the definition with a detailed discussion.
>
> # Typo in Eq 14.
>
> The observation on both fronts are accurate. The min on the left-hand side of the eq. 14 is over both v and u; this was a typo, and the replicated version in the appendix has the corrected variant. We will update this in the final draft.

---

> > ### Comment · Reviewer_k4wz · 2021-08-27
> > **Thank you for the response**
> >
> > Thank you for the message.
> >
> > - I obviously agree that it is infeasible to evaluate the exact marginal likelihood with y in 10^8 dimensions, but in my opinion, it would still be very informative to do the comparison for the largest feasible scale, e.g. 10^4.
> >
> > - About the choice of fonts: I do not have strong opinions about the specific choice of fonts, but sometimes it is crucial to be able to distinguish between the random variable itself and a particular realization of it.
> >
> > After seeing the rebuttal and the other reviews, I still believe that this is a good paper, so my score is unchanged.

---

### Official Review · Reviewer_NXpx · 2021-07-16

**Rating:** 7
**Confidence:** 3

**Summary:**

An approach vor variational inference in large-scale hierarchical models is presented that is more efficient than the base lines. This is achieved using shared parameters (amortization) and a feature pooling network. Experiments confirm the efficiency of the approach on synthetic data as well as the MovieLens dataset.

**Ethical Concerns:**

no issues

**Limitations And Societal Impact:**

yes

**Main Review:**

The paper is original and clearly written (all terms are defined, although some only later after the first mention).

Regarding the significance my thoughts are as follows:
This paper has a focus on the theory and thus does not spend too much space on the experiments. The theory regarding the branch approximations is interesting and seems to be relevant. What is not entirely clear to me from the paper is how widely applicable the proposed methods are. The cited application papers in the introduction are all older works before the boom of neural network based approaches. In my view the main limitation of the paper is thus an appropriate comparison to a state-of-the-art model (such as VAE maybe, or a NN approach for learning hierarchical models) to show, either theoretically or practically, how the proposed method compares to more recent methods. This would make the results, which are more or less just comparing variants of the same method, much more convincing. In general, though, I think this is a good paper and the above reservations should maybe not be weighted too heavily.

Corrolary -> Corollary

**Time Spent Reviewing:**

3

---

> ### Author Response · Authors · 2021-08-10
> **We thank the reviewer for their comments and offer specific responses below.**
>
>
> Thank you for the appreciation of the content on branch approximation and the typo suggestions.
>
> To clarify some of the concerns, we wish to emphasis that the proposed methods do not immediately extend to variational auto-encoders. The aim of VAE is to learn a generative model via maximizing the lower bound to likelihood; we aim to develop more accurate and scalable black-box inference methods for cases where hierarchical model is given/known. Thus, comparisons to VAEs or an NN approach for learning hierarchical models seems to be on a tangent to our goal of black box inference method.
>
> In retrospect, we should have made the distinction between VAEs and our methods (BBVI) abundantly clear. We propose to make this explicit in the Introduction. In addition, we believe an extended relevant work section addressing the concerns of reviewer F7hF should also add more context and distinction to our paper.

---

### Official Review · Reviewer_qcmy · 2021-07-16

**Rating:** 6
**Confidence:** 2

**Summary:**

This work proposes a Gaussian naive bayes approximate posterior for a 2-level hierarchical naive bayes model. Various parameterizations for the variational posterior are explored, including a fully coupled joint posterior, a factored branch posterior, and an amortized version of the latter. Results and theory indicate that having an approximate posterior that makes the same independence assumptions as the true posterior does not hurt, and that parameter sharing improves performance at scale.

**Limitations And Societal Impact:**

* The appendix includes some discussion of the limitations of the different posterior parameterizations.

**Main Review:**

*Originality*
* It is unclear whether the contribution of this paper, the exploration of the parameterization of the Gaussian naive bayes approximate posterior, is novel. To remedy this, the related work must cover similar models and inference methods. Currently the related work is too high-level, and offers little context. Similar generative models must have been used elsewhere, such as hierarchical naive bayes [1]. Please find and cite the relevant work in order to convince us that the model is useful and that amortized variational inference has not been applied in this exact setting.
* The preference for the branch posterior rather than the fully-connected joint can be motivated by Webb et. al. [2].

*Quality*
* The exploration of the approximate posterior is thorough. However, not enough context is given for experimental results. How do the models perform relative to other baseline models? How is evaluation normally conducted on MovieLens?

*Clarity*
* The paper is very clearly written, and seems technically sound.

*Significance*
* Providing more context will help show the significance of the work. The methods and experiments section are complete, but giving more context for the background and results will complete the paper.

*Comments*
* The title is a bit too high-level and vague as well. This paper proposes amortized variational inference for a very particular model, I suggest including that in the title.

[1] Langseth, H. and Thomas D. Nielsen. “Classification using Hierarchical Naïve Bayes models.” Machine Learning 63 (2006): 135-159.

[2] Stefan Webb, Adam Golinski, Robert Zinkov, N Siddharth, Tom Rainforth, Yee Whye Teh, and Frank Wood. Faithful inversion of generative models for effective amortized inference. In Advances in Neural Information Processing Systems, 2018.

Edited score: After calibrating across other reviews, the issues I raised are either not high priority (experimental) or can easily be fixed (more context), and have lowered confidence accordingly. I would be willing to raise the score higher if those are addressed.

**Time Spent Reviewing:**

2

---

> ### Author Response · Authors · 2021-08-10
> **We thank the reviewer for their comments and offer specific responses below.**
>
> # General Comments
>
> - “This work proposes a Gaussian naive bayes approximate posterior for a 2-level hierarchical naive bayes model….”
>
>     We respectfully believe this is an incorrect characterization of our work. The central idea of naive Bayes methods is to neglect dependencies for tractability. The motivation of our work is precisely not to introduce any extra independencies other than those that hold in the true posterior. The scope of HBD models is beyond naive Bayes. For instance, $y_i$ in eq. 1 can be a multivariate quantity with a complex dependence on $(\theta, z_i, x_i)$. Moreover, we do not propose to use a naive Bayes approximate posterior; the aim of the paper is to develop scalable inference approaches that are as accurate as a fully joint posterior and yet orders of magnitude more scalable.
>
> - “It is unclear whether the contribution of this paper, the exploration of the parameterization of the Gaussian naive bayes approximate posterior, is novel…”
>
>     We respectfully disagree that our paper explores the parameterization of the Gaussian naive Bayes approximate posterior. While our experiments are based on a Gaussian approximation, the most widely used approximation for BBVI, the theory we provide is independent of the family of $q$. Further, again, our approximate posterior provably introduces no independencies that do not hold in the true posterior.
>
>     In general, we demonstrate that amortized branch approaches for a joint approximation could be similarly accurate and be more scalable.
>
> - "...convince us that the model is useful and that amortized variational inference has not been applied in this exact setting."
>
>     We wish to iterate that we do not propose any particular generative model; instead, we propose an inference method for a general class of models (that are not learned as part of our framework). Moreover, we believe that our work is the first instance of a scalable yet accurate amortized black-box inference method for hierarchical models. We will be obliged if the reviewer can point us to papers where the amortized black-box variational inference has been conducted for non-conjugate hierarchical models; otherwise, the claim of "must have been used elsewhere" and "find and cite it" seem unsubstantiated.
>
> # Context
>
> We agree that an extended discussion of the context will benefit the paper and believe the discussion on the papers cited by the R#F7hF should help establish that. Please, refer to the detailed comments offered to R#F7hF.
>
> Also, thank you for pointing us towards the paper from Webb et al. 2018, and we will add the citation appropriately.
>
> # Reconsidering the title.
>
> We considered using a different title but noticed in the literature that the "hierarchical model" is used for the same case [1, 2] so we aimed for consistency. Nevertheless, we are open to changing the title if that would be better. We welcome suggestions; we propose "Amortized black-box variational inference for hierarchical branch distributions” to make it more specific.
>
> # References
>
> 1. Hoffman, Matthew D., et al. "Stochastic variational inference." Journal of Machine Learning Research (2013).
> 2. Matthew D. Hoffman and David M. Blei. Stochastic structured variational inference. AISTATS, 2015.

---

> > ### Comment · Reviewer_qcmy · 2021-08-10
> > **Reviewer response**
> >
> > NB: Naive bayes does not preclude complex dependencies, only conditional independence of the children given the parent. I did miss that $y_i$ can depend on $\theta$, which does mean NB is an incorrect classification of the generative model.
> >
> > Usefulness of the generative model: I would like to be convinced that the generative model (HBD) is important, otherwise how else could you motivate working on scaling inference for this particular generative model? Ideally HBD would have been used elsewhere, for similar reasons. This is not a comment on the novelty of the inference procedure, but rather a request for more context on the motivation for the problem itself (i.e. the generative model / HBD). My main issue is that the introduction and related work seem a bit disconnected from the specific generative model itself: which of the related works are exact instantiations of the generative model of focus? For the related work that were exact instantiations of the generative model of interest, is there a speed comparison of the proposed inference method with prior inference methods? If all the cited works are exact instantiations of the generative model of interest, then further experiments demonstrating the generality of the approach would be helpful. I will be happy to change my score if this is a misunderstanding.
> >
> > Title: I like this more specific title.

---

> > > ### Author Response · Authors · 2021-08-28
> > > **We thank the reviewer for enaging in the discussion, and provide specific comments below.**
> > >
> > > ### Naive Bayes
> > >
> > > We believe we were just unfamiliar with the reviewer's usage of the term Naive Bayes. Overall, we are satisfied that there is no longer a confusion in the structure of the dependencies, and that it was just a terminological issue.
> > >
> > > ### Examples of the models
> > >
> > > There are numerous places that the model in Fig. 1 is used. Hierarchical linear regression, hierarchical probit regression, and hierarchical logistic regression are among the most commonly occurring hierarchical models that fit into the definition of HBDs. If we consider Stan—the leading probabilistic framework—as a metric for real world usage, hierarchical models make up for a good majority of use-cases [12, 13]; HBDs are one of the simplest hierarchical models.
> > >
> > > Additionally, the methods we discuss apply to hierarchical models that may have some discrete variables in addition to the HBD structure, such as topic model, mixture models, and hidden Markov Models (see [sec. 2, 9] for discussion on marginalization for models with discrete parameters.)
> > >
> > > Moreover, we do not impose conjugacy at any level. Thus, in addition to existing models in mainstream, our methods will benefits practitioners in carrying out previously untested hypothesis across the fields where hierarchical models are used.
> > >
> > > We propose to add this discussion to establish more context for HBDs and for applicability of our methods.
> > >
> > > ### Existing algorithms for HBD.
> > >
> > > We want to emphasis that all existing algorithms for this model are either black-box (e.g. work on non-conjugate models [10, 9, 11]) or amortized (scale to large datasets [3-8]), but none are both. In keeping with our motivation of black-box models, we compared to non-amortized black-box methods.
> > >
> > > For non-amortized BBVI baselines, we use Gaussian VI ($q_{\phi}^{\text{Joint}}$ in our notation); this is by far the most common black-box VI variational family [10, 9, 11]; we experiment with three variants of it: diagonal, block diagonal, and dense covariance (see Fig. 5 and 6). The structured variants ($q_{v,w}^{\text{Branch}}$ in our notation) are the natural extension of the BBVI baselines (as they exploit structure and reduce the number of parameters—while not compromising the accuracy (from Thm 2).) The overall aim of our experiments is thus to demonstrate that Amortized VI not only performs similar to such non-amortized BBVI baselines, but converges faster and scales better (see fig. 6).
> > >
> > > Moreover, our methods apply for non-conjugate models. Other methods [1, 2, 8] for hierarchical models rely on simplifying assumptions, and are not applicable for such non-conjugate models. For conjugate models, existing approaches use natural gradients [1, 2, 8, and several others] and we do not expect better performance when compared to these methods (we use regular gradients); in contrast, our approach is the first scalable BBVI method for non-conjugate hierarchical models.
> > >
> > > We propose to add this discussion to the paper to establish more context and hope this remedies the reviewer's concerns.
> > >
> > > Finally, if another black-box method that applies to large non-conjugate HBD models can be pointed out, we would be happy to compare to it. To the best of our knowledge no such algorithm exists.
> > >
> > > **References**
> > >
> > > 1. Hoffman, Matthew D., et al. "Stochastic variational inference." Journal of Machine Learning Research (2013).
> > > 2. Matthew D. Hoffman and David M. Blei. Stochastic structured variational inference. AISTATS, 2015.
> > > 3. Edwards, Harrison, and Amos Storkey. "Towards a neural statistician." ICLR, 2017.
> > > 4. Lee, Dong Bok, et al. "Meta-GMVAE: Mixture of Gaussian VAE for
> > > Unsupervised Meta-Learning." ICLR, 2020.
> > > 5. Esmaeili, Babak, et al. "Structured neural topic models for
> > > reviews." AISTATS, 2019.
> > > 6. Ilse, Maximilian, et al. "Diva: Domain invariant variational
> > > autoencoders." Workshop on Deep Generative Models for Highly Structured
> > > Data, ICLR, 2019.
> > > 7. Bouchacourt, Diane, Ryota Tomioka, and Sebastian Nowozin.
> > > "Multi-level variational autoencoder: Learning disentangled
> > > representations from grouped observations." AAAI, 2018.
> > > 8. Johnson, Matthew J., et al. "Composing graphical models with
> > > neural networks for structured representations and fast inference."
> > > NIPS, 2016.
> > > 9. Kucukelbir, Alp, et al. "Automatic differentiation variational inference." The Journal of Machine Learning Research 18.1 (2017): 430-474.
> > > 10. Rajesh Ranganath, Sean Gerrish, and David Blei. Black Box Variational Inference. In AISTATS, 2014.
> > > 11. Abhinav Agrawal, Daniel R. Sheldon, and Justin Domke. Advances in black-box VI: normalizing flows, importance weighting, and optimization. In NeurIPS, 2020.
> > > 12. Stan Developers.Example Models, 2018. URLhttps://github.com/stan-dev/example-models.
> > > 13. Stan Development Team.The Stan Core Library, Version 2.18.0., 2018. URLhttp://mc-stan.org.

---

### Official Review · Reviewer_F7hF · 2021-07-16

**Rating:** 5
**Confidence:** 4

**Summary:**

This paper makes an attempt to address the problem of large-scale inference for hierarchical probabilistic graphical models that contain both global and local latent variables by proposing to use Amortized VI instead of VI in [1] for more efficient and scalable inference. The authors first define a general class of hierarchical models which they call Hierarchical Branch Distributions (HBDs). They then propose a variational family matching this class of models for performing amortized inference. The authors then perform a series of experiments on synthetic datasets and the MoveLens dataset where they evaluate different inference models in terms of the final ELBO and convergence.

**Limitations And Societal Impact:**

To the best of my knowledge, this work does not raise any major social impact.

**Main Review:**

[General Comment]

In my opinion, the main weakness of the current manuscript is the lack of novelty or at least the lack of transparency of the contributions. While I very appreciate the importance of accurate/scalable inference for hierarchical graphical models and therefore am a big fan of this line of research, I found it very difficult to pin-point the exact contributions of this paper. Amortized variational inference has been applied not only to models with solely local variables, but also models with global and local variables as well, so this is not a new idea by any means. Here are just a few examples: [2-6] that all use AVI and most contain local and global variables. In fact, [2] also uses pooling to extract group content from local features. So if the main contribution can be described as Figure 3 (‘d’ in particular), it’s hard for me to see the novelty here as I would classify that as basically choosing an appropriate conditional variational family for your model. Ofcourse if this was an application paper, where an extensive set of experiments had been done  with the specified model to a particular task to achieve good results, it would have been a different story but this is not the case here.



[Clarity]

I did not find the paper easy to read, which I believe also partially contributed to the contributions not being very clear. I would encourage the authors to proofread the paper once more. I also do not find all the mathematical statements clear or necessary. For example, the “symmetric“ definition in page 2 is redundant as far as I can see (a=b ⇒ p(a|c) = p(b|c)). Theorem 2 (& paragraph 4 in page 1) is also obvious and redundant I would argue. Choosing a variational family that factorizes just your target would of course yield a more accurate inference.

-There is a font switch for theta on page 3 (compare equation 4 & 5)
-Typo in equation (16)



[Novelty & Prior work]

I don't think the current manuscript did a good job of discussing related work. See my general comment for more details.
Another missing reference is [7]


[Minor comments]

In order to properly test the inference, I also would advise the authors to test the model containing no ‘x’. While the authors state that “results extend easily to case where there are no local covariates”, this is not necessarily the case in practice depending on  how much information the covariates provide about ‘y’. In such cases, the model does not have to rely on the latent variables so accurate inference might not matter as much to achieve a good likelihood.




[References]

[1] Hoffman, Matthew D., et al. "Stochastic variational inference." Journal of Machine Learning Research 14.5 (2013).

[2] Edwards, Harrison, and Amos Storkey. "Towards a neural statistician." arXiv preprint arXiv:1606.02185 (2016).

[3] Lee, Dong Bok, et al. "Meta-GMVAE: Mixture of Gaussian VAE for Unsupervised Meta-Learning." International Conference on Learning Representations. 2020.

[4] Esmaeili, Babak, et al. "Structured neural topic models for reviews." The 22nd International Conference on Artificial Intelligence and Statistics. PMLR, 2019.

[5] Ilse, Maximilian, et al. "Diva: Domain invariant variational autoencoders." Medical Imaging with Deep Learning. PMLR, 2020.

[6] Bouchacourt, Diane, Ryota Tomioka, and Sebastian Nowozin. "Multi-level variational autoencoder: Learning disentangled representations from grouped observations." Thirty-Second AAAI Conference on Artificial Intelligence. 2018.

[7] Johnson, Matthew J., et al. "Composing graphical models with neural networks for structured representations and fast inference." Advances in neural information processing systems 29 (2016): 2946-2954.

**Time Spent Reviewing:**

6 + 2 (rebuttal)

---

> ### Author Response · Authors · 2021-08-10
> **We thank the reviewer for their detailed comments. We start the discussion by summarizing our contributions and then offer specific comments for other concerns.**
>
>
> # Contributions
>
> - We present the first high-accuracy black box VI algorithm that scales to large hierarchical branch models. To the best of our knowledge, all previous approaches (including all those mentioned in the review) missed this in at least one of: (1) not applying to general "black-box" models; (2) using mean-field inference which is far less accurate (as we demonstrate) or (3) being far less scalable (e.g. using a full-rank Gaussian). Our approach is as accurate as using a full-rank Gaussian yet scales as well as amortized mean-field methods and does not require the existence of any specialized expectations. No previous work achieves this.
> - While we certainly agree that theorem 2 is intuitive, it is critical to the justification of our method, as it is used in the proof of claim 5, and has a non-trivial proof.
> - In section 5, our analysis gives way to a natural amortization scheme. To achieve similar accuracy as joint approximations, we can amortize the function from local data points $(x_i, y_i)$ to parameters $w_i$ of the local distributions $q_{w_i}(z_i \vert \theta)$. Note, this significantly reduces the architecture space. For instance, without our analysis, it is unclear how global latent variables $\theta$ should be conditioned on for amortized distributions.
> - In section 6, we propose an encoder architecture that allows for permutation invariance and arbitrary sized local inputs. In section 7, we present experimental validation for our main claim ”similarly accurate black box inference that scales to models that are orders of magnitude larger.”
>
>
>
> # Context
>
> ### Neural Statistician [3]
>
> Thank you for pointing out this paper. We agree that the construction of the statistics network is similar to our feature network, and it was an oversight not to discuss this work in our submission. However, we want to draw attention to the differences in the two setups.
>
> First, the two line of work differ fundamentally: we provide BBVI methods for HBDs, whereas the aim of Neural Statistician is to learn representations of related datasets for downstream tasks. Second, HBDs have global variables $\theta,$ whereas there are only local variables (at the dataset or the data point level) in the Neural Statistician model. It is not immediately obvious how the neural statistician model extends to setting where global latent variables are present; our analysis in Section 5, outputs a straightforward amortization scheme in the presence of $\theta$, drastically reducing the architecture space while still allowing for accurate inference.
>
> Overall, we agree that the feature network can be viewed as a variant of the statistics network and propose to cite the paper and offer adequate discussion to demonstrate their precedence.
>
> ### Other works with hierarchical models and amortization [3-7].
>
> Again, thank you, for bringing these works to our attention. We want to emphasize that none of these works are aimed at black box inference—our main focus. All these approaches apply hierarchical models to a specific problem, and the main aim is to learn a better generative model. This often leads to specialized assumptions on the class of distributions [4, 5], the structure of dependence [3, 6], or the approximate inference network [Sec 3.3, 7].
>
> Further, in all of the mentioned works, there are no global parameters equivalent of $\theta$ in an HBD, for which there is a Bayesian treatment. It is not trivial how these approaches will be extended to a global setting. As described above (comment on  “Neural Statistician”), it is unclear how exactly to extend the amortization scheme under the presence of a global variable; our analysis in section 5, uncovers a straightforward method for amortization in such scenarios.
>
>
> ### Structured VAE [8]
>
> Again, thank you for bringing this paper to our attention and will add it to the paper with adequate discussion. Overall, the aim in SVAE is still to learn structured generative models. In SVAE, authors use conjugacy in addition to amortization for local distributions. Further, SVAE does not consider the case where local observations are a collection of iid observation. In comparison, we do not impose any conjugacy restrictions on our models and provide a general purpose, scalable yet accurate inference strategy for a broad class of models.
> We propose to include this work into our relevant works section with adequate discussion on differences.
>
> Overall, we thank the reviewer for pointing out the missing references and propose to add the above discussions to help establish more context.
>
> # Clarity
>
> ### Fonts for random variables
>
> We use the sans-serif font to represent random variables explicitly (we note this in L90 before the first use.) This is a fairly common notational choice, which we believe is superior to the common convention in NeurIPS papers to have random variables with no notational marking at all. Nevertheless, as this was an issue for more than one reviewer, we acknowledge that this is something that should be fixed. We propose to emphasize this more prominently in the paper. We would also appreciate further feedback on this issue: For example, would it be better to change to a bold sans-serif font for random variables?
>
> The random variable font—sans serif—is used when dealing with expectations or properties of random variables; for instance, in eq. 6 we want to emphasis $\hat {\mathcal L}$ is a stochastic estimator. It is not used when a variable is just an argument to a function or an arbitrary value, as this is not a random variable (e.g. eq. 1 - eq. 3.)
>
> ### “Symmetric” definition
>
> The gist is that for a general hierarchical branch models, $p(y_1 \vert \theta, z_1, x_1)$ and $p(y_2 \vert \theta, z_2, x_2)$ may not necessarily be in the same class of distributions. In doing this, we employ a rather common abuse of notation where $p(a)$ denotes the distribution of $a$, and the functional form of $p$ is different for different variables. The definition in page 2 emphasizes that in a symmetric model, $p(y_1 \vert \theta, z_1, x_1)$ and $p(y_2 \vert \theta, z_2, x_2)$ are necessarily in the same distribution family, and if the conditioning variables take the same value then the distributions will be identical.
>
> Citing this as an important concern, we propose to add the above discussion and make the overload of notation more explicit.
>
> ### Theorem 2
>
> We respectfully disagree that theorem 2 is obvious or redundant. First, note that $p$ in theorem 2 is any HBD and not necessarily symmetric. Second, the theorem works for any joint variational approximation $q(\theta, z)$ (again, this need not be symmetric).
>
> This theorem is essential for our later results (it is used in claim 5). The guarantee that our algorithm is as accurate as using a full-rank distribution would not be possible without this result.
>
> We agree the result is extremely natural (also stated in L114-115, and L28-L30 that theorem "confirms intuition"); however, the proof (section B of the supplement) is non-trivial, despite great effort being devoted to simplify it. If there is a claim that the theorem is obvious, we would be obliged if the reviewer could sketch a simpler/shorter proof to validate that claim. Similarly, if there is a claim of redundancy, it would be helpful to sketch how this theorem could be skipped.
>
>
>
>
>
>
> # References
>
> 1. Hoffman, Matthew D., et al. "Stochastic variational inference." Journal of Machine Learning Research (2013).
> 2. Matthew D. Hoffman and David M. Blei. Stochastic structured variational inference. AISTATS, 2015.
> 3. Edwards, Harrison, and Amos Storkey. "Towards a neural statistician." ICLR, 2017.
> 4. Lee, Dong Bok, et al. "Meta-GMVAE: Mixture of Gaussian VAE for
> Unsupervised Meta-Learning." ICLR, 2020.
> 5. Esmaeili, Babak, et al. "Structured neural topic models for
> reviews." AISTATS, 2019.
> 6. Ilse, Maximilian, et al. "Diva: Domain invariant variational autoencoders." Workshop on Deep Generative Models for Highly Structured Data, ICLR, 2019.
> 7. Bouchacourt, Diane, Ryota Tomioka, and Sebastian Nowozin.
> "Multi-level variational autoencoder: Learning disentangled
> representations from grouped observations." AAAI, 2018.
> 8. Johnson, Matthew J., et al. "Composing graphical models with
> neural networks for structured representations and fast inference."
> NIPS, 2016.

---

> > ### Author Response · Authors · 2021-08-27
> > **Dear reviewer, did we address your concerns?**
> >
> > Dear reviewer,
> >
> > We were hoping you would have a chance to consider our response. Has it addressed your concerns? If not, we would value the opportunity to further discuss any issues that you felt were still of concern.

---

> > > ### Comment · Reviewer_F7hF · 2021-08-28
> > > **Thank you for your response, and apologies for the delay.**
> > >
> > > Thank you very much for your comments, and my deepest apologies for joining the discussion late.
> > >
> > > I have read the response as well as other reviews. Overall, while I have re-considered some of my original criticism, I unfortunately must say I am not fully convinced by the rest of response particularly with regards to novelty/significance.
> > >
> > > **Theorem 2 And choice of q**
> > >
> > > After re-reading the paper abit more carefully one more time, I believe I originally slightly misunderstood theorem 2. I was originally was thinking of the following decomposition of the posterior $p(\theta|z)\prod_ip(z_i|y_i)$ with the matching $q(\theta|z)\prod_iq(z_i|y_i)$. However, in this paper, the inference distribution $q$ decomposes in the same direction as the generative model instead of its inverse model, which I think makes the theorem not as intuitive as I thought and more interesting (in most deep generative models, the q is simply a (faithful) inversion of p). That being said, the theorem still make sense since it sill matches to just a different decomposition of the generative model. Moreover, I completely agree with the authors about the proof not being trivial so I'm happy to withdraw this point.
> > >
> > > **Notation**
> > >
> > > Thank you for the clarification! I now understand the notational choice. My background is more on the deep generative model side so I had not seen this notation before, therefore I thought it might be simply a typo. But I'm aware that some of the statistics paper also use the same notation. While now I think the current notation is fine, I think using bold sans-serif is definitely a good solution.
> > >
> > > **Significance and related work**
> > >
> > > - "Black-box models": I am not sure I understand the claim that the other referenced papers are "not applying to general "black-box" models". How is this also not true for this paper? We know everything about $p$ and we designed a $q$ to match that $p$. Furthermore, while those papers do consider specific models (similar to this paper), it doesn't mean that the ideas for the model design as well hoe they compute the ELBO cannot be easily extended to similar models.
> > >
> > > - "Black-box VI": I'm also not sure about the statement "we provide BBVI methods for HBDs". Was a reinforce-style gradient used to train the objective here? Sorry but I could find this on the paper. While I'm not familiar with the method used in [25], STL is definitely not a BBVI method as it uses reparameterization.
> > >
> > > - Furthermore, just because other papers such as neural statistician used reparameterization, it doesn't mean that other types of gradient estimations cannot also be applied to those models. This choice is simply because reparameterization works best for high-dimensional data (with deep neural nets) while BBVI is very unstable for such cases.
> > >
> > > - Incorporating $\theta$: While the response states that considering a global variable $\theta$ for other models such a neural statistician would not be straightforward, I fail to see why this is the case. Neural statistician paper presents a method for inferring global variables from local variables (going from the inner plate to the outer plate). As far as I can see, simply adding another plate should not cause any additional problem besides adding $\log p(\theta)/q(\theta)$ to the objective. Also please note that the experiments in some of these papers are on high dimensional images and the parameters to be optimized are conv-nets which is more difficult than the settings considered in this paper.
> > >
> > > I have increased my score from to a weak reject but due to the reasons stated above, I am still hesitant to recommend acceptance.
> > >
> > > Best

---

> > > > ### Author Response · Authors · 2021-08-31
> > > > **We thank the reviewer for engaging in the discussion, and offer specific points below.**
> > > >
> > > > We believe that many of these comments are the result of a miscommunication stemming from our use of the term "black-box" in our discussion here. We apologize for this: in the paper itself we were careful in describing our scope [e.g. lines 69-85]; we emphasize that the term "black-box" does not appear in the paper (except for in citations) and we hope that the paper can be evaluated in that light.
> > > >
> > > > ### Black box models.
> > > >
> > > > > "Black-box models": I am not sure I understand the claim that the other referenced papers are "not applying to general "black-box" models". How is this also not true for this paper? We know everything about and we designed a to match that . Furthermore, while those papers do consider specific models (similar to this paper), it doesn't mean that the ideas for the model design as well hoe they compute the ELBO cannot be easily extended to similar models.
> > > >
> > > > We were using the term "black-box" to refer to a setting where inference can only access $\log p$ or/and $\nabla_{z} \log p$ [1, 2, 3] (or parts of it). So, the model itself is treated as a black-box in the sense that the algorithm has no access to $p$ other than the ability to evaluate it or its gradient for individual terms in the factorization, and thus can apply to any distribution with the factorization. This is in contrast to, e.g., conjugate methods where it is assumed that (parts of) $q$ and $p$ are simple enough that expectations can be computed [4, 5, 8]. Thus, we do not know *everything* about $p$—we do not know how to compute any expectations over $q$ of any parts of $p$.
> > > >
> > > > The motivation for this line of work is the "probabilistic programming" setting, where a use will choose a dataset and a model $p$ and then ask the system to perform inference on it. In this setting it is important to be able to handle as large a class of target models $p$ as possible, so that users have flexibility in choosing the model that suits their application.
> > > >
> > > > In comparison, deep generative models aim to learn a particular model for particular application. This is a very different setting, since there is less of a boundary between a "user" who might design the model and the "system" that would do inference on it. Thus, the deep generative models community (sensibly) places much less focus on the importance of black-box algorithms.
> > > >
> > > > ### Black-box VI
> > > >
> > > > > "Black-box VI": I'm also not sure about the statement "we provide BBVI methods for HBDs". Was a reinforce-style gradient used to train the objective here? Sorry but I could find this on the paper. While I'm not familiar with the method used in [25], STL is definitely not a BBVI method as it uses reparameterization.
> > > >
> > > > We did not use a reinforce-style gradient estimator. We used a the re-parameterization gradient estimator referred to as the "total gradient" in the STL paper.  We suspect the reviewer is referring to the reinforce gradient of the Ranganathan et. al. [1] when they introduced the term "black-box VI". Since that time, it has become common in the inference field to use the term "black-box" inference in a more general way for inference methods that can apply to more general target distributions (without simplifying assumptions like conjugacy) [2, 3, 6, 7, and others]. So black-box VI methods often use gradient estimators like STL [3, 6, 7]. In fact, using the right estimator for BBVI is a whole problem in itself [7].
> > > >
> > > > > Furthermore, just because other papers such as neural statistician used reparameterization, it doesn't mean that other types of gradient estimations cannot also be applied to those models. This choice is simply because reparameterization works best for high-dimensional data (with deep neural nets) while BBVI is very unstable for such cases.
> > > >
> > > > We think this concern is addressed by the earlier clarifications.
> > > >
> > > > ### Incorporating $\theta$
> > > >
> > > > > Incorporating θ: While the response states that considering a global variable θ for other models such a neural statistician would not be straightforward, I fail to see why this is the case. Neural statistician paper presents a method for inferring global variables from local variables (going from the inner plate to the outer plate). As far as I can see, simply adding another plate should not cause any additional problem besides adding log⁡ p(θ)/q(θ) to the objective. Also please note that the experiments in some of these papers are on high dimensional images and the parameters to be optimized are conv-nets which is more difficult than the settings considered in this paper.
> > > >
> > > > The central issue is that the variational distribution must reflect the dependence structure between the local variables $z_i$ and the global variables $\theta$. To see why it is not obvious to extend the neural statistician to models with $\theta$, note that the overall approximation is given by $q(\theta, z \vert x, y) = q(\theta \vert x, y) \prod_{i=1}^{n} q(z_i \vert \theta, x_i, y_i)$. We use a parameterized distribution for $q_v(\theta)$ and amortize for $q(z_i \vert \theta, x_i, y_i)$. There are non-trivial design choices for the construction of the corresponding amortization function. The point of Theorem 2 and Claim 5 is to derive exactly what form $q(z_i \vert \theta, x_i, y_i)$ must take to ensure equal accuracy as if one had simply chosen a given joint distribution $q(z_1, \cdots, z_N, \theta \vert x_i, y_i)$.
> > > >
> > > > For specific example, consider the conditional Gaussian expression in the Appendix D, Eq. 31. Do you amortize with $(\theta, x_i, y_i)$ to output $(\mu_i, A_i, \Sigma_i)$? Do you still reuse $\theta$ as in Eq. 31? If $\Sigma_i$ depends on $\theta$ (it would if you amortize with $(\theta, x_i, y_i)$), will this still be equivalent to doing fully-joint Gaussian VI?
> > > >
> > > > In general, what decision will you make for some other conditional distribution? In section 5, our analysis answers all such questions and lays down the foundation for how to amortize in such cases.
> > > >
> > > > Thus, adding $\theta$ is not as simple as adding a $\log (p(\theta) / q(\theta))$ term unless one is willing to make harmful assumptions, e.g. that $\theta$ is independent of $z_i$ in the variational posterior.
> > > >
> > > > ### References
> > > >
> > > > 1. Rajesh Ranganath, Sean Gerrish, and David Blei. Black Box Variational Inference. In AISTATS, 2014.
> > > > 2. Kucukelbir, Alp, et al. "Automatic differentiation variational inference." The Journal of Machine Learning Research 18.1 (2017): 430-474.
> > > > 3. Abhinav Agrawal, Daniel R. Sheldon, and Justin Domke. Advances in black-box VI: normalizing flows, importance weighting, and optimization. In NeurIPS, 2020.
> > > > 4. Hoffman, Matthew D., et al. "Stochastic variational inference." Journal of Machine Learning Research (2013).
> > > > 5. Matthew D. Hoffman and David M. Blei. Stochastic structured variational inference. AISTATS, 2015.
> > > > 6. Hoffman, Matthew, and Yian Ma. "Black-Box Variational Inference as a Parametric Approximation to Langevin Dynamics." International Conference on Machine Learning. PMLR, 2020.
> > > > 7. Geffner, Tomas and Justin Domke. “A Rule for Gradient Estimator Selection, with an Application to Variational Inference.” AISTATS (2020).
> > > > 8. Johnson, Matthew J., et al. "Composing graphical models with
> > > > neural networks for structured representations and fast inference."
> > > > NIPS, 2016.

---

### Decision · Program_Chairs · 2021-09-27

**Decision:**

Accept (Poster)

**Comment:**

The authors consider variational inference for a hierarchical model where each observation is generated using a local/observation-specific latent variable, which in turn depends on the global latent variable. They prove that using a variational posterior with the same dependence structure as the model results in inference accuracy comparable to a general joint posterior approximation at a much lower computational cost and allows observation subsampling for an additional speedup. Experiments confirm the theoretical results and show that amortizing local inference enables training on much larger datasets without affecting accuracy.

The paper describes a simple but effective idea clearly and shows that it works in practice. The discussion of related work is insufficient however, and should be extended to include the Neural Statistician and its descendants and their relationship to this work. Also, as pointed out by a reviewer, the title of the paper is too general and should really be made more specific, as the method applies only to hierarchical models with a very specific, if common structure.